# The Role of Exposomes in the Pathophysiology of Autoimmune Diseases II: Pathogens

Aristo Vojdani [1,2,*] , Elroy Vojdani [3], Avi Z. Rosenberg [4] and Yehuda Shoenfeld [5,6,7,*]

1   Immunosciences Lab, Inc., Los Angeles, CA 90035, USA
2   Cyrex Laboratories, LLC, Phoenix, AZ 85034, USA
3   Regenera Medical, Los Angeles, CA 90025, USA; evojdani@gmail.com
4   Department of Pathology, Johns Hopkins University, Baltimore, MD 21218, USA; avirosenberg@gmail.com
5   Zabludowicz Center for Autoimmune Diseases, Sheba Medical Center, Tel-Aviv University, Ramat Gan 5265601, Israel
6   School of Medicine, Ariel University, Ariel 4076414, Israel
7   Laboratory of the Mosaic of Autoimmunity, Saint Petersburg State University, 199034 Saint Petersburg, Russia
*   Correspondence: drari@msn.com (A.V.); yehuda_shoenfeld@sheba.health.gov.il (Y.S.)

**Abstract:** In our continuing examination of the role of exposomes in autoimmune disease, we use this review to focus on pathogens. Infections are major contributors to the pathophysiology of autoimmune diseases through various mechanisms, foremost being molecular mimicry, when the structural similarity between the pathogen and a human tissue antigen leads to autoimmune reactivity and even autoimmune disease. The three best examples of this are oral pathogens, SARS-CoV-2, and the herpesviruses. Oral pathogens reach the gut, disturb the microbiota, increase gut permeability, cause local inflammation, and generate autoantigens, leading to systemic inflammation, multiple autoimmune reactivities, and systemic autoimmunity. The COVID-19 pandemic put the spotlight on SARS-CoV-2, which has been called "the autoimmune virus." We explore in detail the evidence supporting this. We also describe how viruses, in particular herpesviruses, have a role in the induction of many different autoimmune diseases, detailing the various mechanisms involved. Lastly, we discuss the microbiome and the beneficial microbiota that populate it. We look at the role of the gut microbiome in autoimmune disorders, because of its role in regulating the immune system. Dysbiosis of the microbiota in the gut microbiome can lead to multiple autoimmune disorders. We conclude that understanding the precise roles and relationships shared by all these factors that comprise the exposome and identifying early events and root causes of these disorders can help us to develop more targeted therapeutic protocols for the management of this worldwide epidemic of autoimmunity.

**Keywords:** exposome; pathophysiology; autoimmunity; environmental trigger; pathogen; infection; autoantibodies; molecular mimicry; bystander activation; epitope spreading

## 1. Introduction

In our previous publication on the involvement of exposomes in autoimmune diseases we focused on toxic chemicals and food [1]. To briefly reprise what was covered in Part 1 of this series, the exposome has been described as "life-course environmental exposures (including lifestyle factors) from the prenatal period onwards" [2]. It was also described more recently as "the science of social-to-biological transitions" [3]. In plain language, the exposome is an individual's accumulated lifetime exposure to both external and internal environmental factors, which includes food/diet, toxic chemicals, infectious pathogens, and lifestyle.

Toxic chemicals are ubiquitously prevalent in the modern world and can directly or indirectly damage our tissues and organs, cause the release of autoantigens and the formation of neoantigens, and thence lead to autoimmunity [1]. In much the same way, the food that we eat every day commonly has colorants, preservatives, taste-enhancers,

or packaging that can contribute chemical contamination [1]. Food has the added aspect of inducing sensitivity or allergies in individuals disposed to react to certain foods [1]. While the focus on food in Part 1 was as an exposure factor, at this point it must be pointed out that, of course, apart from its nutrient value, there are exposures to food that may be beneficial towards disease risk [4,5].

As the previous article has already discussed the environmental factors of food and chemicals, this current review examines the role of pathogens. This brings us to the infectome, a concept introduced by Bogdanos et al. in 2013 to denote the part of the exposome referring to the collection of an individual's exposure to infectious agents [6]. Described by Bogdanos et al. as a platform to trace infectious triggers of autoimmunity, the infectome directly relates to geoepidemiological, serological, and molecular evidence of the co-occurrence of several infectious agents associated with autoimmune diseases that may provide clues as to the root causes or triggers of autoimmunity. Infectious agents are among the major environmental factors that contribute to many autoimmune diseases (ADs). One classic example is beta hemolytic streptococcus; infection with this pathogen leads to rheumatic fever several weeks later. The structural similarity or molecular mimicry between the bacterial M5 protein and human $\alpha$-myosin can cause the production of autoantibodies against the host's own $\alpha$-myosin in susceptible individuals, potentially leading to autoimmunity [7–10].

There are actually different mechanisms by which autoimmunity can be induced by infectious agents such as viruses, bacteria, parasites, and fungi. Often the induction of autoimmunity is the result, not of one single infection, but rather a "burden of infections" stemming from childhood [11]. This interaction between a variety of different mechanisms, different environmental triggers, different infections, and different autoimmune diseases means that there are a myriad of possibilities of mechanisms and relationships between different infectious species and types of autoimmunity. Some of these possibilities were reviewed by the corresponding author in a previous study [12], which showed that more than 20 infectious agents in patients with rheumatoid arthritis (RA) and more than 10 infectious agents in patients with thyroid autoimmunity played a role in the pathophysiology of autoimmune disease.

Infectious agents can induce autoimmune disorders through the following mechanisms [11]:

(1)    Molecular mimicry
(2)    Epitope spreading
(3)    Viral persistence
(4)    Bystander activation
(5)    Polyclonal activation
(6)    Autoinflammatory activation of innate immunity
(7)    Dysregulation of immune homeostasis

Although infections may not always be directly responsible for the induction of autoimmunity, they can sometimes target the sites of autoimmune inflammation, thus affecting the autoimmune disorder in one of three ways: (a) exacerbating the ongoing disease, which leads to greater severity and duration, (b) inducing a relapse; (c) leading to chronic progressive disease [13].

As early as 2000, Blank, Krause, and Shoenfeld described various examples of molecular mimicry in different autoimmune diseases, detailing how viruses, microbes and parasites could break peripheral tolerance and induce or maintain autoimmunity through several overlapping mechanisms, and how the matching of synthetic peptides with corresponding autoantigens may be used for treatment [14]. Abu-Shakra et al. also presented evidence for a role for molecular mimicry between host and parasites in autoimmunity [15], while Kanduc and Shoenfeld showed the high degree of peptide sharing between human papillomavirus epitopes and human proteins [16]. Dotan et al. showed how molecular mimicry between homologous peptides in severe acute respiratory syndrome coronavirus 2 (SARS-CoV-2) and numerous fertility-linked proteins of the female reproductive system

might result in the development of autoantibodies and the onset of related autoimmune manifestations [17].

This review focuses mainly on molecular mimicry or antigenic mimicry, the most likely mechanism by which infection induces autoimmunity. This is because foreign antigens often have significant structural similarity to self-antigens. An immune response to microbial antigens could therefore result in the activation of T cells that cross-react with these self-antigens, because a single T cell can react to different peptides that have similar charge distributions and a generally similar shape [11,17]. Table 1 shows a sampling of viral or bacterial antigens, their cross-reactivity with different self-antigens, and possible resulting autoimmune diseases.

**Table 1.** Viral and bacterial antigens, their cross-reactive self-antigens, and possibly resulting diseases.

| Pathogen Antigen | Cross-Reactive Self-Antigens | Autoimmune Disease |
|---|---|---|
| Herpes simplex virus | Corneal antigen | Stromal keratitis |
| Campylobacter jejuni | Ganglioside in peripheral nerve | Guillain-Barré syndrome |
| Coxsackievirus | Glutamic acid decarboxylase | Type 1 diabetes |
| Theiler's murine encephalomyelitis virus | Proteolipid protein | Multiple sclerosis |
| Yersinia enterocolitica | Thyrotropin receptor | Thyroid autoimmunity |
| Borrelia burgdorferi | Leukocyte function associated antigen | Lyme arthritis |
| Salmonella typhi and Yersinia enterocolitica | HLA-B27 | Reactive arthritis |
| HHV-6, EBV, Rubeolla, influenza virus, and HPV | Myelin basic protein | Multiple sclerosis |
| Streptococcal M protein | Myosin and other heart valve proteins | Rheumatic fever |
| Porphyromonas gingivalis | Heat-shock proteins | Atherosclerosis |
| Trypanosoma cruzi | Cardiac myosis | Chagas heart disease |
| SARS-CoV-2 | More than 20 tissue antigens | More than 20 ADs |

As an example, the immune system might launch an immune response against a foreign virus, such as coxsackievirus, which shares amino acid chains with host proteins such as glutamic acid decarboxylase 65 (GAD-65). The immune response may produce a cross-reactive antibody that mistakenly identifies host GAD-65 as a foreign antigen. The generated antibodies might then attack GAD-65 proteins, leading to tissue damage and perhaps to type 1 diabetes [18].

The great numbers of different microbial proteins and the ways they could cross-react with human proteins means that immune response against microbial antigens will not always result in autoimmune disease. But the initial immune response could still result in epitope spreading, or the exposure of other regions of the same self-protein and the production of additional antibodies against it. In 1997, Craft and Fatenejad described epitope spreading as the autoantibody response diversifying to other components via recognition of new epitopes within the intact complex [19]. In later years other researchers such as James, Sercarz, and Monneaux have well established the contribution of epitope spreading to autoimmunity [20–22]. In 2009 Kivity et al. [11] reviewed and summarized the criteria for the mechanism by which autoimmunity is induced.

In this review, we will use oral pathogens, SARS-CoV-2, and herpesviruses as three of the best examples of bacterial or viral induction of autoimmunity.

## 2. Oral Pathogens and Autoimmunity

Periodontitis is a major inflammatory disease of the gums induced by bacterial infection. Although the human subgingival plaque harbors more than 500 bacterial species [23], for decades researchers believed that *Aggregatibacter actinomycetemcomitans* was the most likely etiologic agent in aggressive periodontitis [24]. However, later research pointed to *Porphyromonas gingivalis* as the primary oral pathogen [25–27]. Pathophysiologically, this organism is associated with autoimmune diseases, cardio-metabolic disorders, Alzheimer's disease, and cancer. The potential link between this bacteria and associated disorder is supported by experimental animal studies, and the fact that biomarkers which are detected in

comorbid conditions are ameliorated by local treatment of periodontitis [26,27]. Although even more recent research from 2020 now identifies the so-called "Red Complex" trio of *P. gingivalis*, *Treponema denticola*, and *Tannerella forsythia* as the species highly associated with severe periodontal disease [28], the following section focuses on *P. gingivalis* alone for certain unique features described below that associate it with autoimmunity.

*P. gingivalis* is also associated with different disorders and areas distant from the oral cavity, such as atheromatous plaques, amyloid plaques, the gut, and the joints, mainly due to the protease gingipain, which is important not only for the colonization of subgingival tooth sites, but also for the presence and severity of *P. gingivalis* in extra-oral sites [26]. In the case of colitis, the pathobionts can reach the gut as a result of swallowing, where, in genetically susceptible individuals, colitis can be promoted through the induction of proinflammatory cytokine production by inflammatory macrophages and other cells. The gingipain proteases disrupt endothelial barriers and increase permeability, degrading platelet endothelial cell adhesion molecules, inducing the production of proinflammatory cytokines, and triggering platelet aggregation, potentially inducing or exacerbating atherogenesis.

Additionally, Th17 cells reactive to oral pathobionts, after their expansion during periodontitis, migrate to the gut via the lymphatic system. Phagocytosis of colonized oral pathogens by antigen-presenting cells (APCs) and the release of IL-1β contributes to the activation and proliferation of orally migrated Th17 in the gut. The Th17 cells produce IL-17 and other cytokines, exacerbating intestinal inflammation and resulting in colitis [26].

*P. gingivalis* also expresses a very unique enzyme, peptidyl-arginine-deiminase (PAD), which cleaves proteins and exposes their C-terminal arginine residues for citrullination in joints and other tissues, and the production of antibodies against citrullinated proteins (Figure 1) [29–35]. Enzymes such as this have a key role in autoimmune diseases such as celiac disease and RA; this is why RA can cause inflammation not just in joints but also other organs, such as the heart, lungs, skin, and peripheral nerves, often with grave consequences [36]. PAD2 and PAD4 are the most strongly associated PAD enzymes in RA [37]. Anti-PAD2 antibodies are implicated with less severe joint and lung disease in RA patients; anti-PAD4 antibodies are associated with severe bone damage. Dysregulated PAD enzyme function and the development of anti-citrullinated protein antibodies can promote three important features of RA: citrullination, the production of inflammatory cytokines, and bone destruction.

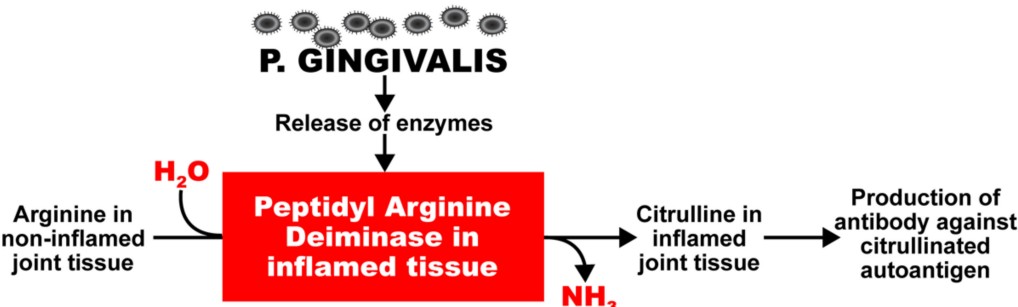

**Figure 1.** Potentiation by oral pathogens. Oral pathogens such as *P. gingivalis* can potentiate the deamination of arginine or formation of citrullinated proteins and peptides in joint and other tissues.

The above is a classic example of how environmental factors can change a self-tissue antigen into a disease-associated T-cell epitope, leading to the generation of antibodies against the citrulline-containing new epitope. If celiac disease as an autoimmune disorder has been shown to be driven by transglutaminse-2 and deamidated gliadin, we can then state that RA is caused by environmental triggers, such as *P. gingivalis* or Epstein–Barr virus (EBV), because these triggers or factors cause the formation of various citrullinated self-epitopes, such as α-enolase, fibrin, collagen type II, filaggrin, vimentin, and keratin [30–38]. In this mechanism, the simultaneous citrullination of self-antigens and bacterial proteins

generates the structure of a neoepitope. The neoepitope can cause a breakdown in the body's self-tolerance, causing antibodies to be produced against both the citrullinated bacterial antigens and the citrullinated self-proteins. One of these bacterial antigens is α-enolase, which shares a significant homology with human α-enolase. Thus, antibodies produced against the bacterial α-enolase will also attack the human α-enolase, and vice versa. This may explain why elevated levels of α-enolase antibodies are detected in the synovium of 60% of patients suffering from RA [33].

In fact, the homology between α-enolase and *P. gingivalis* has been documented in a library of cyclic citrullinated α-enolase peptides. Using this library, immunological mapping was able to identify a B-cell-dominant epitope that included AA 5–21 of α-enolase or KIHAREIFDSRGNPTVE that had an 82% similarity with the sequence of *P. gingivalis*, and in which arginine-9 and arginine-15 were citrullinated [39,40]. It has been shown that immunization with citrullinated human α-enolase, citrullinated *P. gingivalis* α-enolase, and citrullinated fibrinogen results in similar pathology in humanized DR4 transgenic mice. This mechanism just described may well be a common factor linking cardiovascular disease and autoimmunity. These findings indicate that through mimicry of the structural sequence of host-citrullinated proteins, *P. gingivalis* peptidylarginine deiminase-citrullinated bacterial α-enolase could cause a breakdown in the tolerance of structurally similar host proteins, inducing the production of anti-citrullinated protein antibodies and the development of RA [41]. In a majority of patients, the antibodies thus generated can be detected up to 14 years before the actual clinical onset of RA and the production of IgM antibodies against IgG (rheumatoid factor) [42].

The specificity of anti-citrullinated peptide in the joint is enhanced through epitope spreading to other citrullinated autoantigens, such as collagen, filaggrin, fibrinogen, and vimentin (see Figure 2).

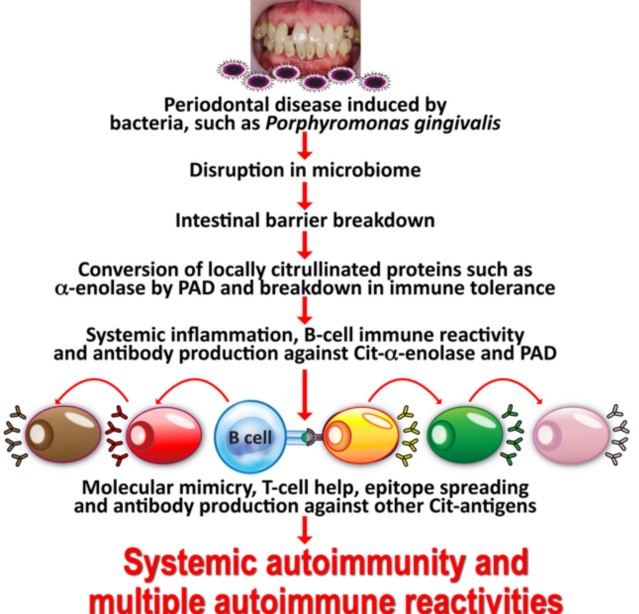

**Figure 2.** Mechanism by which oral infection can lead to multiple autoimmune reactivities. Oral infectious agents reach the gut and disturb the gut flora, leading to a disrupted epithelium and gut permeability. Local inflammation generates autoantigens by PAD; antibody production starts against one autoantigen, but through epitope spreading reaches out to multiple antigens, leading to systemic inflammation, multiple autoimmune reactivities, and systemic autoimmunity.

Interestingly, we recently measured citrullinated antibodies against α-enolase and many additional tissue antigens in the blood of patients suffering from COVID-19 with different degrees of severity and found that antibody levels were significantly elevated

in patients compared with healthy controls [43]. Our findings further support the notion that SARS-CoV-2 infection contributes to the production of autoimmune antibodies, as has been claimed recently in other articles, including some by one of this manuscript's co-authors [44,45].

Taking all the aforementioned information into consideration, one can infer that different multiple autoimmune reactivities share some commonalities in the mechanisms of their immunopathogenesis. For one, environmental factors such as xenobiotics can induce the formation of neoantigens in genetically susceptible individuals. In this same subgroup, *P. gingivalis* can induce the citrullination of host self-proteins and convert them to autoantigens. The immune system can then mistakenly identify these converted self-proteins as foreign or harmful pathogens/antigens, producing antibodies against the body's own tissues, thus triggering the inflammatory process involved in the clinical manifestations at the heart of autoimmune disorders.

### 3. SARS-CoV-2: The Autoimmune Virus

Since the outbreak of coronavirus disease 2019 (COVID-19), evidence has accumulated over the course of the past two years showing the involvement of SARS-CoV-2 infection in immune dysregulation and autoimmunity [44–55]. In individuals, presumably genetically pre-disposed, the reported symptoms related to inflammation and autoimmunity correlate with the detection of certain circulating inflammatory mediators and autoantibodies that support the diagnosis of various autoimmune disorders in a subgroup of patients with SARS-CoV-2 [44,45,49,50,56–62].

Autoimmune diseases can be identified and classified by certain characteristics, such as the detection of autoantibodies. The loss of immune tolerance and dysregulation of the immune system can lead to long-lasting inflammatory reactions, as well as malfunction and damage of target organs [63]. These immune-mediated afflictions are also found in COVID-19. SARS-CoV-2 infection induces immune reactions, which may have very important ramifications in the development of treatments and vaccines against this virus [52,64,65].

In patients with COVID-19, the infiltration and activation of immune cells play a role in the pathogenesis of organ injury. Macrophage activation syndrome (MAS) is increasingly being recognized as part of the continuum of cytokine storm syndrome, especially the production of IL-6; Conti et al. asserted that this could lead to potentially life-threatening complications in COVID-19 [66]. In 2020 Wampler Muskardin noted that, in MAS, activated macrophages produce an excessive amount of proinflammatory cytokines, subsequently polarizing into the inflammatory M1 phenotype and exhibiting cytotoxic dysfunction [67].

As part of the cellular and humoral immune responses, T cell immunity plays a central role in the control of SARS-CoV-2 infection. Neutralizing antibody responses and the antigen-specific T cell subsets CD4 and CD8 play defensive roles against SARS-CoV-2, while dysfunctional or impaired immune responses, such as a deficiency of native Tcells, may lead to undesirable disease outcomes [65].

Tests performed in clinical laboratories have documented the effects of both lymphopenia and lymphocytosis on T cells, B cells, CD4, CD8, Th1, Th2, Th17, Treg and natural killer cells [66–70]. Both lymphocytosis and lymphopenia, especially the latter's effect on lymphocyte subset distribution, have been associated with the severity and mortality of COVID-19. Other conditions found to be associated with COVID-19 are excessive IL-6 production, neutrophilia, and associated excessive neutrophil extracellular traps, which paralleled lung injury in patients with severe COVID-19 [71–73].

It can be seen that certain events and environmental factors can influence the functioning and efficiency of the immune system's component cells, and this can make an immune response a double-edged sword that can both protect and do harm, depending on the state of the immune cells and cytokines. In COVID-19, excessive production of normally beneficial cytokines has been associated with disease severity. Damage-associated molecular patterns (DAMPs) have also been shown to participate in COVID-19's pathogenesis and disease outcome, similar to their roles in autoimmune diseases [71,72,74,75]. The

activation of extrafollicular B cells was found in critically ill COVID-19 patients, similar to that observed in autoimmunity, correlating with the production of high levels of SARS-CoV-2-specific neutralizing antibodies and poor disease outcome [76]. Patients with COVID-19 showed changes in their peripheral blood B cell subpopulations, with atypical memory B cells expanding, while classical memory B cells decreased [77]. Severe COVID-19 patients showed greater proportions of mature natural killer (NK) cells, and reduced proportions of T cells [78].

Different studies in which deep immunophenotyping was performed have found three different patterns or lymphocyte maps in COVID-19 patients, some with decreases, some with increases, and some with no changes observed in their lymphocyte subpopulations [78–82]. These various patterns of lymphocyte subpopulations have very significant implications in how to treat patients with COVID-19 and other viral infections, since patients with lymphopenia and lymphocytosis will react differently to immune modifiers. However, currently, regardless of their lymphocyte pattern or map, patients who are severely and critically ill with COVID-19 and have shown a robust immune response to it are being given immunomodulatory drugs and biological agents that target pro-inflammatory cytokines. These medications have been used for years to target autoimmune disorders [83,84]. Understanding the relationships shared by different lymphocyte maps, immune responses, and the severity of COVID-19 in patients will be extremely helpful in formulating the proper therapeutic protocols for the different stages of this modern-day pandemic [78,85–90]. Furthermore, close monitoring of the corresponding T cell subsets that mimic those already studied with autoimmune disorders can provide invaluable information on a patient's recovery progress or changes in his condition during treatment [81–84].

SARS-CoV-2 has been called the autoimmune virus, and the following key points support this appellation:

1.  Similarities in lymphocyte map or lymphocyte subpopulation patterns between COVID-19 and autoimmune diseases
2.  Molecular mimicry between SARS-CoV-2 spike proteins, nucleoproteins and human autoantigens that contribute to autoimmune diseases
3.  Reaction of both animal and human monoclonal antibodies made against SARS-CoV-2 spike proteins and nucleoproteins with human autoantigens
4.  Reaction of antibodies made against human autoantigens with SARS-CoV-2 spike proteins and nucleoproteins
5.  Detection of autoantibodies made against human autoantigens known to cross-react with SARS-CoV-2 in the sera of patients with COVID-19

These five key points are the best experimental evidence for SARS-CoV-2 being the autoimmune virus.

### 3.1. Keypoint No. 1: Similarities in Lymphocyte Map or Lymphocyte Subpopulation Patterns between COVID-19 and Autoimmune Disorders

Our body's immune defense system is an intricate compilation of different kinds of cells that work together to protect their host from pathogenic invaders (see Figure 3). These various cells must combine in just the right numbers, percentages, ratios, and proportions in order to achieve immune balance, the state when all the cells are in the optimum working ratios, the immune defenses are working correctly, and the host body is healthy. These cells are the basic soldiers and foundation of the body's immune defense.

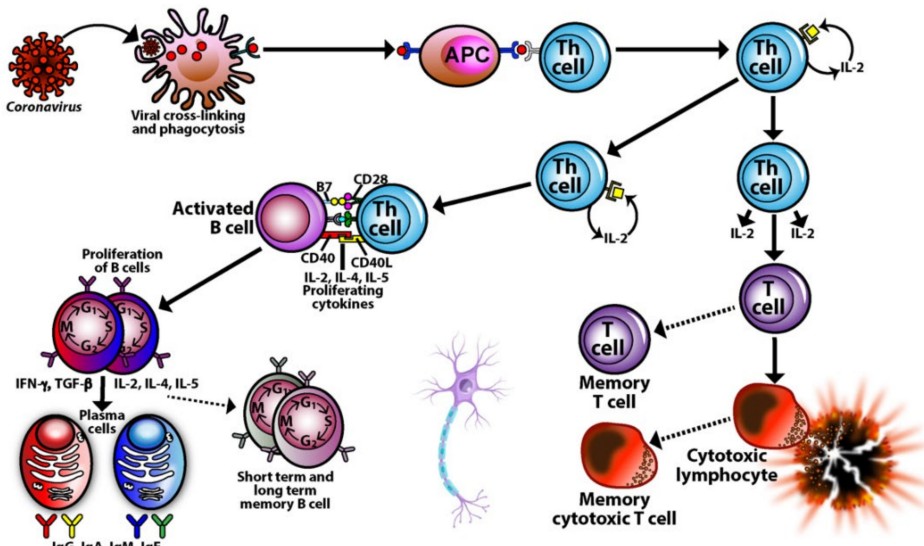

**Figure 3.** The interactive network of the immune system. The immune system is a complex network of different kinds of cells that must work together and combine in just the right numbers, ratios, and percentages to protect the host from pathogens.

Lymphocytes and their subpopulations are key elements of the immune defense system, accounting for 20–49% of the total number of white blood cells (WBCs) in adults. Imbalances or improper ratios in these lymphocyte subpopulations could lead to autoimmune disorders [71,80–86]. Autoimmune diseases are marked by both cellular and humoral immune reactions that involve CD4⁺, Th1, Th17, and NK cells. These disorders are also marked by auto-inflammatory reactions due to dysregulated regulatory T cells (Tregs) that lead to organ damage and malfunction [55,84,91–93].

As early as 1992, our own lab was using flow cytometry to perform lymphocyte subpopulation testing on the blood samples of patients with confirmed exposure to a variety of environmental factors, including solvent and breast implants. Even with just normal CBC and blood chemistry testing, one third of these individuals showed abnormalities in T cell, B cell, T helper, T suppressor, CD4/CD8 ratios, and natural killer cell numbers [94–96]. These findings supported the importance of measurements of lymphocyte subsets in patients with a history of exposure to various environmental factors, and with symptoms associated with autoimmune disorders.

Since those early days there have been many advances in the field of immunology and the identification of cluster of differentiation (CD) markers for the cells listed above and more [81,91,97,98]. This makes it easier to use accurate measurements to determine if these cells are indeed working together in the proper balances and ratios to achieve immune harmony, or if they are imbalanced and have fallen into a dominant or deficient immunotype.

For instance, a person who is Th1-dominant is more likely to suffer from inflammation and cellular-mediated autoimmunity. On the other hand, a person with Th2 dominance is more likely to have environmental allergies, hypersensitivity, atopic dermatitis, and antibody-mediated autoimmunity [99–102]. A deficiency in Th17 cells may lead to bacterial and fungal infections, while an over-abundance of Th17 cells may lead to tissue inflammation, systemic autoimmune diseases, and organ-specific autoimmune diseases [103,104]. Treg cells act as conductors or policemen, either conducting the immune orchestra or directing traffic as needed. A deficiency of Tregs with simultaneous elevations in Th1 and Th17 is actually the most pathogenic combination for autoimmunity [105–107]. Conversely, low Tregs with elevated Th2 and Th17 is the most pathogenic combination for allergies and asthma [108,109]. NK, cytotoxic NK, and NKT cells in the right numbers and ratios are important for immunoregulation and protection of the body against viruses and even cancer,

but in the wrong amounts and proportions they may participate in the pathophysiology of autoimmune diseases, COPD, infertility, and even pregnancy losses [110–115].

Thus, it is not enough to just measure the numbers of and ratios of T cells, B cells, CD4 and CD8 cells. It is also important to measure the numbers and calculate the ratios of Th1/Th2 cells and Th17/Treg cells, since imbalances or the wrong proportions in these T-helper subtypes have been observed in a number of autoimmune diseases, such as systemic lupus erythematosus (SLE) [116–118]. Normally, Th1 and Th2 cells regulate and inhibit each other's activities to maintain immune balance. A decrease in Th1 function and simultaneous increase in Th2 function can lead to an over-activation of B cells and the generation of autoantibodies against the body's own tissues, causing tissue damage. As for Th17 cells and Tregs, studies have shown that the ratio between these two types of cells correlate with disease activity in SLE patients. Thus, the monitoring of these biomarkers are very important determinants for disease activity, treatment, follow-up, and recovery for patients with SLE and many other autoimmune diseases [116–118].

Patients with thyroid-directed autoimmunity have demonstrated significant elevations in Th1, normal or low Th2 numbers, and high IgG antibodies to thyroid stimulating hormone receptor (TSHR). Patients with Graves' disease who were treated with anti-thyroid medication showed a significant decrease in the percentage of Th1 lymphocytes 12–24 weeks after treatment, while the percentage of Th2 lymphocytes showed a significant increase. Both the decrease and increase correlated significantly with the patients' clinical conditions [119–121].

In multiple sclerosis (MS), both Th1 and Th17 cells are known to be involved in the pathogenesis of this brain and spinal cord disease. High frequencies of Th1 and Th17 cells correlating with MRI activity have been detected in the blood of patients with MS, showing a dominant contribution of these cells towards clinical disease activity. These cells somehow cross the blood-brain barrier, after which Th1 cells preferentially migrate into the spinal cord, while Th17 cells migrate into the brain, mediating the pathogenesis of MS. This provides strong support for more specific and earlier use of therapy targeting Th1 and Th17 in MS patients. Some MS patients who have been treated with such selected targeting using biologics have remained free of clinical relapse [122–126].

In systemic sclerosis (SSc), patients suffering from the disease have shown an increase in the frequency of Th17 and a decrease in the percentage of Tregs, resulting in an elevation of the Th17/Treg ratio. The number of Th17 cells and their expansion correlated closely with disease activity. Using medication or biologics to neutralize the IL-17 produced by Th17 has been shown to reduce the production of collagen, its deposition in various organs, and autoimmune reactivity in a majority of SSc patients [127,128].

Among the many autoimmune diseases, RA is one in which Th17 plays a particularly important role. In patients with RA, the immune system, particularly Th17, attacks different antigens in the body's joints, resulting in inflammatory symptoms and signs such as pain and swelling, ultimately causing structural damage to the cartilage and bone [129,130]. Flow cytometry tests for RA patients showed an increase in the percentage of Th17 cells and a decrease in the percentage of Th1 cells that correlated with disease activity. This is the best evidence for the contribution of Th17 towards RA, and since normally both Th17 and Th1 are increased in autoimmunity, the increase in Th17 while Th1 decreases opens up a new paradigm in the field of autoimmune disease [129–133]. With regards to treatment, multiple studies have documented the successful use of monoclonal antibodies made against Th17 receptors combined with methotrexate and anti-inflammatory agents in reducing the levels of Th17 cells in the blood to control disease activity in patients [134–136]. In light of the above, changes in Th17 biology and increases or decreases in Th17 cell levels are an important factor in the pathogenesis of RA and supports strategies targeting Th17 using biologics or anti-inflammatory treatments even in the early stages of the disease [134–139].

In psoriasis patients, elevated Th1 cells have been detected in the plaque as well as their blood. This elevation correlates with lower numbers of CD4$^+$CD25$^+$ Tregs and their immunoregulatory capacity. It has been shown that CD4+ T cells are participants

in the pathogenesis of psoriasis, and their elevation coupled with a decrease in Tregs results in a significant increase in the CD4/CD8 ratio, an identifying characteristic of many autoimmune diseases. This impairment of immune function in psoriasis also comes with an abundance of Th17 lymphocytes, which means that the Th17/Treg ratio is elevated as well [140–144]. Using anti-TNF-α or anti-IL-17 to treat psoriasis resulted in an improvement in biomarker levels and an amelioration of inflammation and skin lesions [144,145]. These examples from only a few autoimmune diseases clearly show the importance of measuring the absolute numbers and percentages of T cells, B cells, CD4, CD8, Th1, Th2, Th17, Treg, and NK cells in peripheral blood and the calculation of their ratios as an aid in the detection and treatment of many autoimmune disorders.

Most infected individuals with SARS-CoV-2 are asymptomatic, but a very small number of patients develop severe cases of the disease with multiple organ injuries that overlap with clinical manifestations of a variety of autoimmune diseases. Robust immune reactions in the form of hyperactivation of various cells of the immune system participate in the pathophysiology of both COVID-19 and autoimmunity [55]. These facts strengthen the argument for a role for SARS-CoV-2 in some autoimmune diseases.

Several studies have found that infection with SARS-CoV-2 leads to immune dysregulation and loss of immune tolerance [71–75]. The disease' unique pattern of immune dysfunction included immune dysregulation, major increase or decrease in the number of lymphocytes measured in the blood, MAS, and lower or higher absolute count for CD8 suppressor (cytotoxic lymphocytes), Th1, Th2, Th17, Treg, NK cells, and CD19+/CD45+ B-lymphocytes, in comparison to healthy controls. These changes in the absolute numbers and percentages of these lymphocytes were strongly associated with the highest levels of inflammatory biomarkers (IL-1β, IL-6, IL-17, IL-22, CRP, TNF-α) especially in ICU patients [71–75].

To connect these immune abnormalities to the degree of severity of COVID-19, scientists in one study tested 340 individuals with SARS-CoV-2 infection using measurements of T-cell subsets to see if they could be used as predictors of poor prognosis of the disease [81]. Out of 340 COVID-19 patients, 310 were hospitalized for about two weeks, but recovered and were discharged after successful treatment; the other 30 did not survive the disease. The test results showed that abnormal lymphocyte subsets could predict who would survive the disease, and who would not. For instance, baseline T cell subset numbers differed significantly between those who recovered and those who died [81].

The validity of monitoring lymphocyte subsets as a biomarker of disease, including COVID-19, is supported by a growing number of additions to the concerned scientific literature. A recent article [146] compared Th17 and Treg cell function in healthy controls and SARS-CoV-2 patients. Their data showed that COVID-19 patients had an increase in total number of Th17 cells, but had a significant reduction in the number of Treg cells. This resulted in a significant increase in the Th17/Treg ratio, and a concomitant increase and decrease in the cytokines associated respectively with Th17 and Treg cells. The results showed that, in comparison with controls, increased responses of Th17 cells and decreased responses of Treg cells in SARS-CoV-2 patients had a strong relationship with hyperinflammation, lung damage, and the pathogenesis of disease.

In a different study, a unique immunological profile with altered distribution of peripheral blood lymphocytes usually detected in patients with autoimmune disease was found in COVID-19 patients [77]. Using high dimensional cytometry, blood samples from many COVID-19 patients were analyzed, and three different immune patterns or immunotypes were observed [80]. Immunotype 1 showed robust activation of CD4 T cells with pre-activated or exhausted CD8 T cells that was associated with disease activity. Immunotype 2 was associated with less activation of Th1, CD8+, and proinflammatory B cells, while Immunotype 3 lacked obvious activated T and B cell responses [81]. Another study investigated the relationship between NK cell immunotypes and COVID-19. Strong NK cell activation with high expression of perforin was detected in patients with a severe case of the disease [82].

In a recent study by Phetsoupanh et al., abnormalities in innate immune cells and deficiencies in naive T cells and B cells were found in patients who survived acute COVID-19 and developed post-acute COVID syndrome or long COVID (LC) [147].

Overall, the changes in the absolute numbers, percentages, and ratios of T cells, B cells, CD4, CD8, Th1, Th2, Th17, Treg, NK, and NKT cells help to predict the clinical outcome of patients with COVID-19 and autoimmune diseases [80–82]. Close monitoring of the T-cell subsets and the identification of the different patterns or immunotypes might provide useful data regarding the patient's progress and help to design personalized treatment for patients with autoimmune disease or COVID-19.

Thus, lymphocyte immunotyping can not only define the pattern of lymphocyte subpopulations for a patient but may guide clinicians to tailor or develop personalized treatment for each individual patient who suffers from autoimmunity, COVID-19, or both.

### 3.2. Keypoint No. 2: Molecular Mimicry between SARS-CoV-2 Spike Proteins, Nucleoproteins, and Human Autoantigens Contributes to Autoimmune Diseases

Dotan et al. [45] showed in a very recent study homology of primary sequences between human proteins and components of SARS-CoV-2. Cross-reactivity between these two groups of proteins may trigger the production of autoantibodies that could result in the onset of an autoimmune disease, presenting a possible mechanism in the pathophysiology of autoimmunity. Table 2 below shows only 8 out of the 34 matches in Dotan et al.'s study; the matches involve human proteomes that could lead to grave pathological consequences if they were affected so as to be deficient or dysfunctional [45,47].

**Table 2.** Short list (8/34) and description of human proteomes and the peptide sequences they share with SARS-CoV-2 [25].

| Shared Heptapeptide | Human Proteins Sharing Heptapeptides with SARS-CoV-2 |
| --- | --- |
| SSRSSSR | Corneal antigen |
| ALALLLL | Ganglioside in peripheral nerve |
| ALALLLL | Glutamic acid decarboxylase |
| ALALLLL | Proteolipid protein |
| IGAGICA | Thyrotropin receptor |
| TGRLQSL | Leukocyte function associated antigen |
| NASVVNI | HLA-B27 |
| AEGSRGG | More than 20 tissue antigens |

Kanduc and Shoenfeld [51] addressed the issue of peptide-sharing between SARS-CoV-2 spike glycoprotein and lung surfactant-related proteins, finding homology with 13 out of 24 pentapeptides. They proposed that because of this, the immune response following a SARS-CoV-2 infection might lead to cross-reactivity with pulmonary surfactant proteins, leading to SARS-CoV-2-associated lung disease [51]. In another article, the same authors very recently provided convincing arguments for molecular mimicry as a potential mechanism that contributes towards SARS-CoV-2-associated diseases [52].

Lyons-Weiler provided additional evidence for the presence of cross-reactive epitopes between SARS-CoV-2 and human tissue and their contributing role in autoimmunity when he found a high degree of similarity between immunogenic epitopes of SARS-CoV-2 and various human self-tissue proteins, such as adipose tissue, blood, eye, brain, pituitary gland, thyroid gland, gastrointestinal tract, heart muscle, kidney, liver, lung, skeletal muscle, skin, testes, and many other proteins [47]. The corresponding author of this present article confirmed this similarity with a few variations in his own study, which focused on 55 selected human tissue antigens [50]. While overlapping somewhat with the list of antigens used in the Lyons-Weiler study, we made as the basis for our study key target human tissue proteins that had been found to be involved not just with common autoimmune diseases but also with extra-pulmonary manifestations of COVID-19. We investigated the cross-reactivity of SARS-CoV-2 specifically with specific brain

tissue antigens (α-synuclein, amyloid-β, MBP, NFP, tTG-6, synapsin), M2 protein, liver microsomal peptide, PDH peptide, and specific skin antigens (tTG-2, tTG-3, epithelial cell antigens), while the Lyons-Weiler team studied brain, GI tract, liver, and skin.

We also studied many other tissue antigens, such as barrier proteins, that were not included in the Lyons-Weiler study [50]. We used the Basic Local Alignment Search Tool (BLAST) to determine the degree of identity between SARS-CoV-2 proteins and a carefully selected number of human proteins: F-actin, an important smooth muscle component; thyroid peroxidase (TPO), a target antigen in thyroid autoimmune disorder; and mitochondria M2. We showed that SARS-CoV-2 proteins shared a significant number of peptide sequences with F-actin (58–63%), TPO (50–70%), and mitochondrial M2 protein (50–78%) as shown in Figure 4.

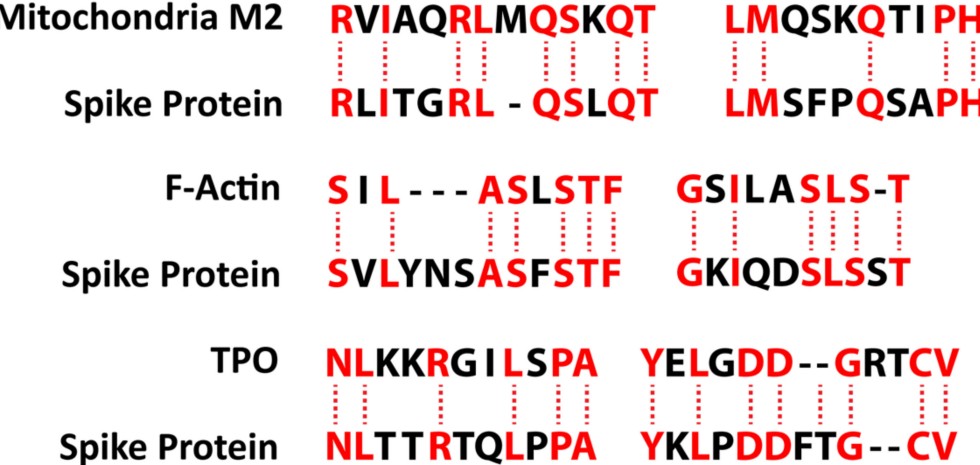

**Figure 4.** Peptide molecular mimicry between SARS-CoV-2 spike protein and mitochondria M2, F-actin, and thyroid peroxidase (TPO) [36].

Furthermore, we found that some peptide sequences had multiple matches with different sections of a SARS-CoV-2 protein. For instance, the F-actin sequence SIL—ASLSTF cross-reacted with the SVLYNSASFSTF sequence in chains A, B, C, and E of the SARS-CoV-2 spike protein, but it also reacted with chain E of the SARS-CoV-2 spike receptor binding domain. We detected many other peptide sequences with identity percentages ranging from 33–49% but they are not shown here.

Such a broad immune cross-reactivity between SARS-CoV-2 proteins and different human autoantigen groups may be involved in the multi-system aspects of the disease, influence its severity, trigger the onset of autoimmunity in susceptible subgroups, and possibly increase the severity of autoimmunity in subjects with pre-existing autoimmune disorders.

*3.3. Keypoint No. 3: Reaction of Both Animal and Human Monoclonal Antibodies Made against SARS-CoV-2 Spike Proteins and Nucleoproteins with Human Tissue Antigens*

In seeking to determine the involvement of SARS-CoV-2 antigens in human autoimmune disorders, our previous study [50] sought to determine whether human monoclonal antibodies that mimic natural antibodies produced against SARS-CoV-2 would react to various human self-antigens. This could be the cause of the multi-system disorder found in patients with severe COVID-19.

In an, earlier, limited study [49], we used mouse monoclonal antibody and rabbit monoclonal antibody made against SARS-CoV-2 proteins to also investigate possible cross-reactivity between SARS-CoV-2 antigens and human autoantigens. We found moderate to strong reactions between these antibodies and 13 different human tissue antigens [49]. While these findings using animal-derived monoclonal antibodies were significant, of course they would be more significant if human-based monoclonal antibody had been

used. Consequently, when human monoclonal antibody to SARS-CoV-2 became available, we applied the antibody to 55 different human tissue antigens detailed in our previously mentioned study [50]. Out of the 55 tissue antigens, the antibody reacted with 28 antigens from different tissue groups, including barrier proteins, skin, muscle, joint, gastrointestinal, thyroid, and neural tissues, and various cellular components (such as ENA and more) [50].

This extensive reactivity of both animal and human monoclonal antibodies to SARS-CoV-2 with so many tissue antigens makes the case even stronger for a possible role for SARS-CoV-2 spike proteins and nucleoproteins in many autoimmune diseases (see Figure 5).

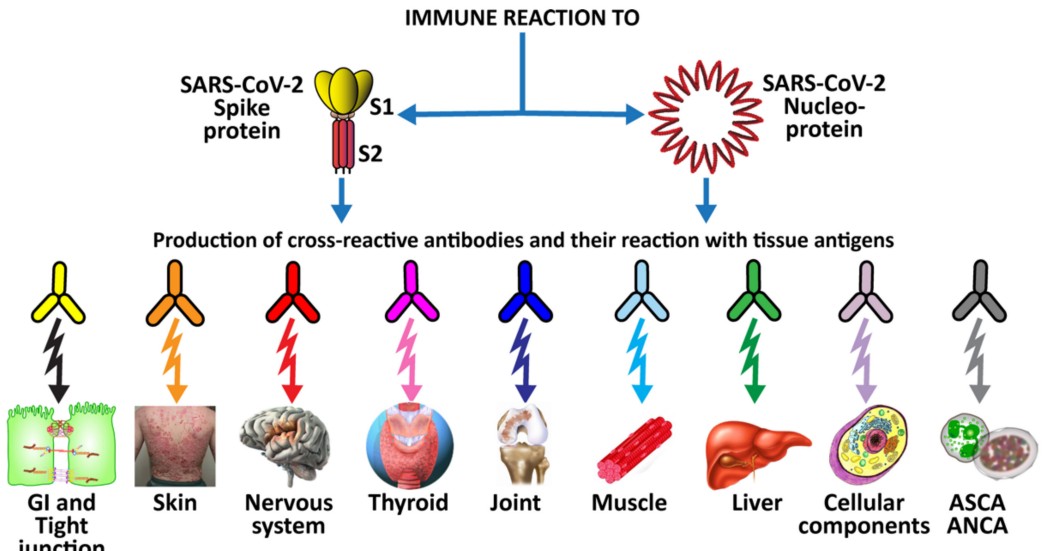

**Figure 5.** Diagram of possible relationship between SARS-CoV-2 proteins and autoimmune target proteins.

*3.4. Keypoint No. 4: Reaction of Antibodies Made against Human Autoantigens with SARS-CoV-2 Spike Proteins and Nucleoproteins*

Mitochondrial M2 antigens share a significant homology or similarity of structure with both SARS-CoV-2 spike proteins and nucleoproteins. In our previous studies [49,50] we found that both animal and human monoclonal antibodies made against both of these SARS-CoV-2 proteins reacted strongly with M2 antigens. We determined this by first using the Trinity Biotech M2 antibody kit to measure the presence of M2 antibody in four control sera and in the serum of four individuals who were positive for M2 antibody. We applied the test kit's negative control calibrator, low and high positive controls, the four negative sera, and the four sera positive for M2 antibody to an ELISA plate coated with both SARS-CoV-2 spike proteins and nucleoproteins. There were moderate reactions from the calibrator and positive controls with known levels of M2 antibody with both spike proteins and nucleoproteins. The four sera with elevated M2 antibody also had moderate reactions with these SARS-CoV-2 proteins. The proteins from the four sera with no detected levels of M2 antibody had no reactions.

This mitochondrial M2 antibody is detected in 90–95% of patients suffering from primary biliary cirrhosis, and sometimes in patients with other liver diseases and scleroderma [49]. Interestingly, a recent study showed that SARS-CoV-2 infection contributes to hepatic impairment in COVID-19 patients [53].

The results of our previous studies support the proposition that molecular mimicry between M2 and SARS-Cov-2 proteins have a role in the production of cross-reactive antibodies. We suggest that similar experiments should be performed with other autoantigens that share homology with SARS-CoV-2 proteins to determine the extent of immunological reactivity between the two groups.

### 3.5. Keypoint No. 5: Detection of Autoantibodies against Human Autoantigens Known to Cross-React with SARS-CoV-2 in the Sera of COVID-19 Patients

The corresponding author participated in a study involving 246 adults from Jewish-American communities across 5 states who had developed symptomatic COVID-19 disease prior to being vaccinated against the disease. Among the participants were 77 randomly selected age- and sex-matched healthy controls who had tested SARS-CoV-2 negative and presented no symptoms of COVID-19. The autoantibody data of these controls were compared to data from 169 individuals who had tested SARS-CoV-2 positive by nasopharyngeal swab and by antibody assay using a kit manufactured by Zeus Scientific. The 169 infected individuals were divided into three groups according to severity of the disease: 74 subjects with fever duration ≤ 1 day and a peak fever temperature of 37.8 °C were designated as COVID-19 mild; 63 subjects with fever duration ≥ 7 days and a peak fever temperature of 38.8 °C were designated as COVID-19 moderate; and 32 subjects presenting severe symptoms and requiring supplemental oxygen therapy were grouped together as COVID-19 severe. This disease severity rating was based on the COVID-19 severity classification from the World Health Organization (WHO). We showed that autoantibodies targeting G protein-coupled receptors and renin-angiotensin system-related molecules associate with the clinical severity of COVID-19 [43].

Using the same 169 sera, in an additional study we measured IgG and IgA antibodies against 58 different tissue antigens representing the brain, heart, lungs, pancreas, liver, skin, muscle, tight junction proteins, blood clotting factors, enzymes, and many cellular components, including nuclear and nucleolar antigens, mitochondria, and double-stranded DNA. In comparison to controls, the COVID-19 mild and moderate groups reacted weakly to 14 out of 58 of the tissue antigens. The COVID-19 severe group showed antibodies at higher levels against 27 tissue antigens for IgG (see Figure 6) and 29 tissue antigens for IgA (see Figure 7).

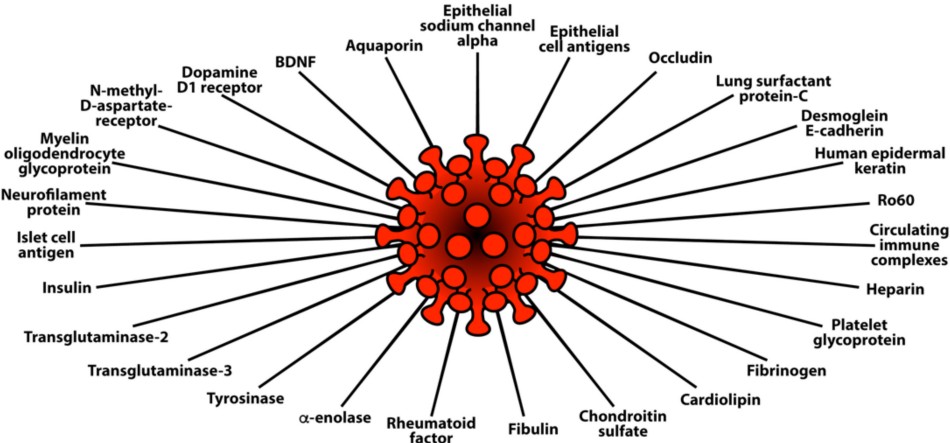

**Figure 6.** Elevations in IgG antibody against 27 different human tissue antigens in the blood of patients with mild to severe SARS-CoV-2 infection (positive for PCR and antibody) in comparison to healthy controls.

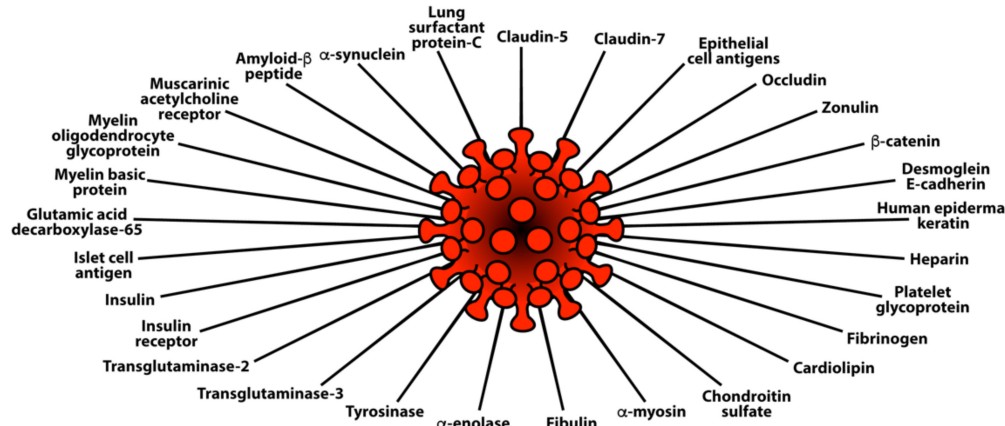

**Figure 7.** Elevations in IgA antibody against 29 different human tissue antigens in the blood of patients with mild to severe SARS-CoV-2 infection (positive for PCR and antibody) in comparison to healthy controls.

Interestingly, there was a significant overlap between tissue-specific antibodies that were detected in patients with mild to severe COVID-19 and tissue antigens that reacted with specific monoclonal antibodies made against SARS-CoV-2 spike proteins and nucleoproteins [49,50].

While these findings of elevated antibodies against various tissue antigens in the blood of COVID-19 patients is supportive of SARS-CoV-2 being the autoimmune virus, affinity purification of these antibodies from COVID-19 patients would be required for further studies on their reaction with these tissue antigens [148].

## 4. Herpesviruses and the Pathophysiology of Autoimmunity

Chronic illnesses in humans are predominantly caused by common viruses. Among them, the most notable ones found to be involved in autoimmune disorders are the family of human herpesviruses: human herpesvirus 1 (HSV-1 or HHV-1); herpes simplex virus type 2 (HSV-2 or HHV-2); the *Varicella zoster* virus (VZV or HHV-3); EBV or HHV-4; cytomegalovirus (CMV or HHV-5); human herpesvirus type 6 (HHV-6); and measles (rubeola) [149,150]. Viruses are generally transmitted through two pathways: horizontal transmission, which is when a susceptible individual becomes infected through close contact with a carrier of the virus through skin, saliva, aerosols, or other body secretions; and vertical transmission, which is when an infected mother passes on the virus to her child during birth, even if the mother is asymptomatic at the time.

As a group, the human herpesvirus family survives and propagates within a host through both acute and persistent infection strategies. Acute viral infection is characterized by rapid onset of disease consisting of a burst of virus replication and what is usually a relatively brief period of mild symptoms that are resolved within days. Think of the common cold. However, in persistent infections, the disease can turn severe, with symptoms becoming alleviated only when specific cell-mediated immune responses isolate and defeat the virus-producing cells. Even then the immune response is not always completely efficient in eliminating the virus from the infected cells, and the human host may become a lifetime carrier of the virus. Thus, persistent infections are characterized by a latent infection in which the virus lies dormant within a cell with alternate cycles of dormancy and activation with gaps of months or even years, with reactivation triggering strong immune responses in the form of IgM and IgG production. Acute and recent infection with a particular virus is marked by high IgM antibody levels, while persistent or chronic infection is marked by high IgG antibody levels against different viral antigens [151–155].

The evidence linking herpesvirus infections to the development of various autoimmune diseases continues to accumulate (see Figure 8). For instance, the pathogenesis of systemic autoimmune diseases (SADs) can be triggered or induced by latent herpesviruses

after reactivation [156–159]. EBV infection is particularly suspected of playing an important part in the pathogenesis of SADs. As evidence of this, studies have found increased EBV viral mRNAs expression and high viral loads of EBV DNA in the blood of SLE patients [160–162]. Patients with RA and Sjögren's syndrome (SS) have also shown high levels of EBV antibodies, increased EBV viral load, and abnormal cell-mediated immunity to EBV [159,163].

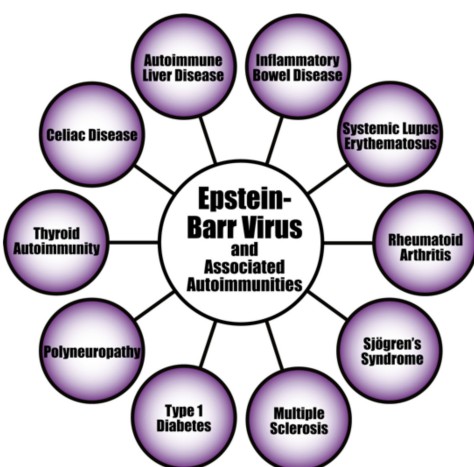

**Figure 8.** Epstein–Barr virus and a few of its associated autoimmune diseases.

Autoimmunity can also be mediated by EBV through molecular mimicry when antibodies produced against EBNA 1 cross-react with autoantigens associated with lupus in SLE patients; subsequent epitope spreading would expand this cross-reactivity to even more autoantigens [164]. EBV infection can also induce the activation of innate immunity through TLR3 signaling, leading to the production of IFN and proinflammatory cytokines [165].

Herpesviruses are neurotropic and neurovirulent, which means they can also infect cells of the central nervous system (CNS), producing neurological illness. Normally the CNS and the eye are immune-privileged sites. This means that the self-antigens against these organs and/or their tissues are protectively segregated from the adaptive immune system, and part of this protective separation is provided by the blood-brain barrier. Unfortunately, environmental factors such as infection can affect this defensive state. Inflammation due to infection at these sites can lead to a disruption in the function of the blood-brain barrier, allowing immunogenic cells to pass through. For instance, studies suggest that susceptibility to MS develops in early childhood, and that viral infections may be the trigger. This means that the herpesviruses infections known to be prevalent in childhood may be triggers of or contribute towards MS [166].

The evidence connecting herpesviruses to autoimmune diseases is very persuasive (see Figure 9). The herpes simplex virus has been found in active plaques from postmortem brain samples taken from MS patients [166]. In fact, HHV-6 has been found to be more prevalent in MS plaques than normal MS white matter, and has been observed to reactivate during MS relapse [167]. EBV has also been linked to MS [168]. HSV has been linked to the autoimmune corneal disease herpetic stromal keratitis (HSK) [169]. In an affirmation of this, the HSV-1-derived protein UL6 was recognized by cornea-specific T cell clones in a murine model due to molecular mimicry [170]. An alternative mechanism for this may be pathogenesis through bystander destruction, because a different study on the involvement of HSV in HSK found that isolated T cells did not cross-react with UL6 [171]. The CMV genome has been reported in type 1 diabetes patients, and persistent CMV infection has been determined to be involved in the production of islet cell antibodies [172,173]. Herpes viral infections have also been connected to the induction of the life-long autoimmune disorder, celiac disease [174,175].

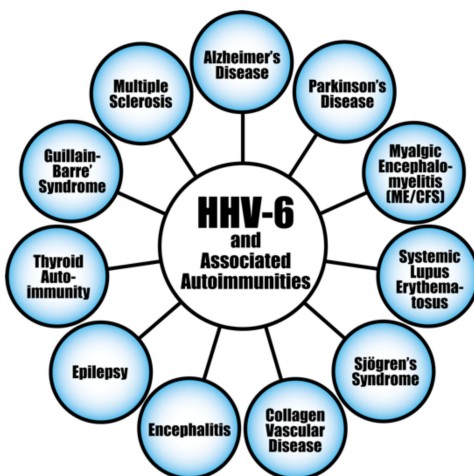

**Figure 9.** HHV-6 and a few of its associated autoimmune diseases.

Finally, viral infection or reactivation with EBV, HHV-6, and other herpesviruses in the CNS can breach the blood-brain barriers, recruiting virus-specific and autoaggressive Th1 and Th17 cells to cause serious damage to neurons [176] (see Figure 10). (1) Herpesviruses enter and infect the body. (2) This causes the activation of T cells and myeloid cells. (3) The activated T cells and myeloid cells penetrate the BBB, which can be weakened by a number of different factors or situations. (4) These cells release IL-1β cytokines, which activate gamma-delta (γδ) T cells. (5) The γδ cells release IL-17, IL-21, and TNF-α, which further activate Th1 and Th17 cells. (6) Activated Th1 cells release IFN-γ and granulocyte-macrophage colony-stimulating factor (GM-CSF), which activates microglia. (7) Activated Th17 cells release IL-17, IFN-γ and TNF-α. (8) The activated Th1 and Th17 cells expand, clone themselves and multiply, and release inflammatory mediators. (9) The released inflammatory mediators, cytotoxic products, and proteases lead to the destruction of neurons' myelin sheath and oligodendrocytes. (10) This leads to the production of antibodies against myelin basic protein (MBP), myelin oligodendrocyte glycoprotein (MOG), and other neuronal antigens.

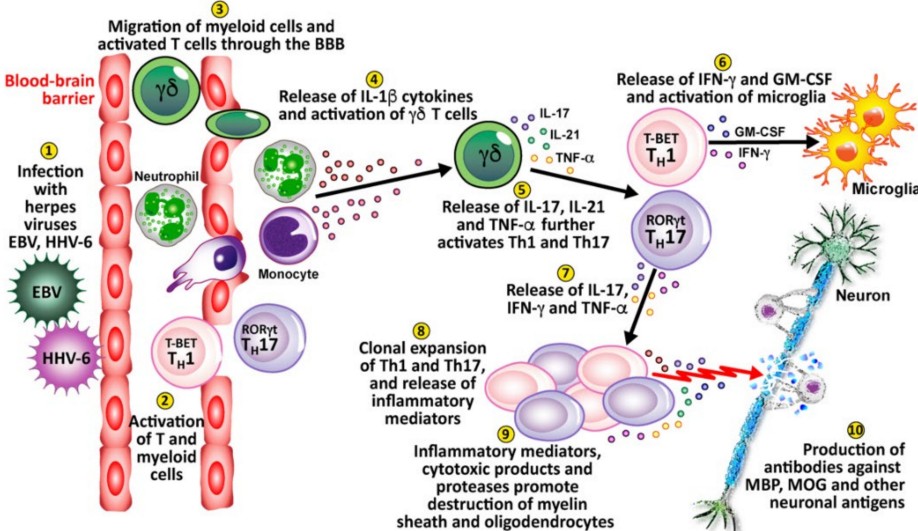

**Figure 10.** 10 key pathological processes induced by the herpes family of viruses that results in the destruction of neurons and the production of antibodies against MBP, MOG, and other neuronal antigens.

*Pathophysiological Mechanisms in the Induction of Autoimmunities by Herpesviruses*

Infections in general, and viruses in particular, are, as a group, considered one of the major environmental factors that can trigger or induce autoimmune disorders in genetically susceptible persons [175,177]. We have shown that herpesviruses are involved in autoimmunity through multiple mechanisms. We have divided these mechanisms into the four that are shown in Figures 11–14: molecular mimicry, bystander activation, disturbed immune surveillance, and epitope spreading.

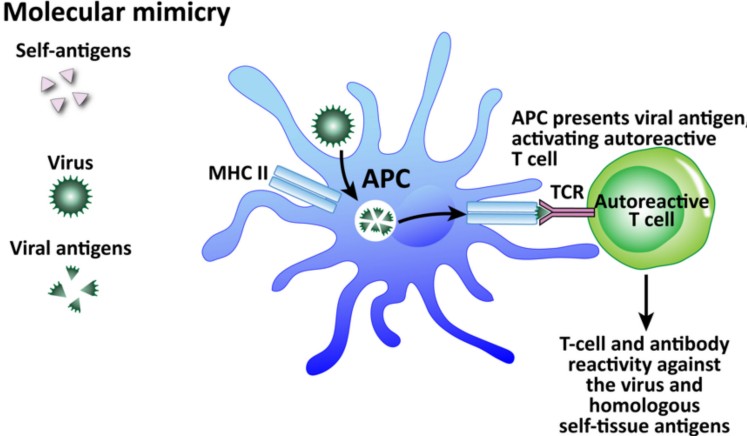

**Figure 11.** How viruses can induce autoimmunity through molecular mimicry. Viruses are taken up by APCs and broken up into viral antigens that are structurally similar to self-antigen. When the APC primes the T cell with the viral antigen, it becomes an autoreactive T cell and attacks both viruses and self-tissue.

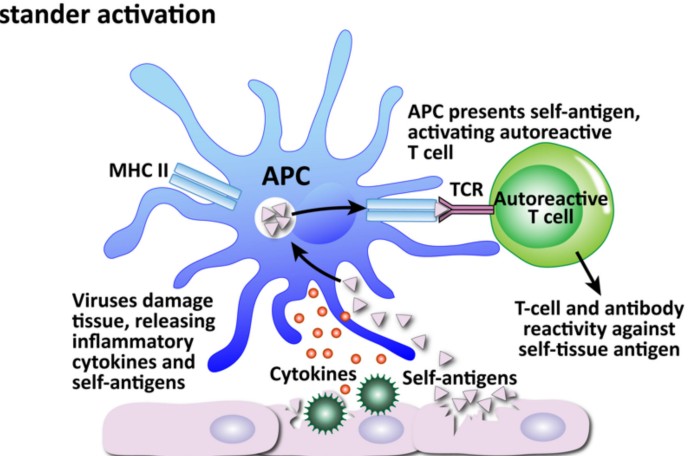

**Figure 12.** How viruses can induce autoimmunity through bystander activation. Excessive non-specific antiviral immune responses lead to the release of inflammatory cytokines and self-antigens. Self-antigens are presented to autoreactive T cells, which attack the innocent "bystander" tissues.

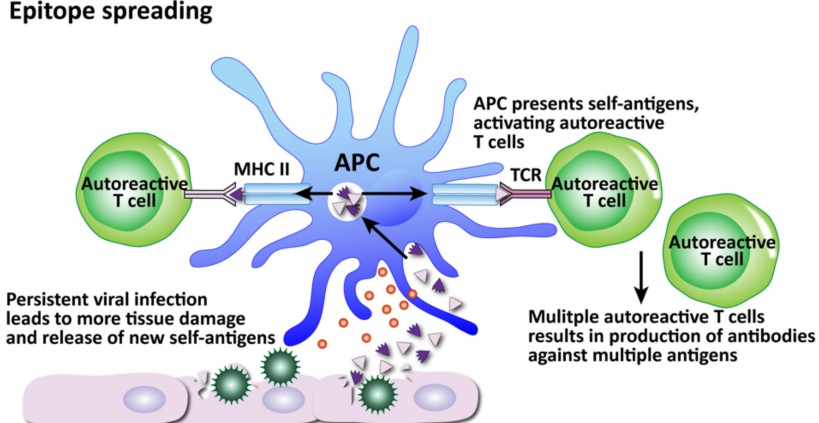

**Figure 13.** How viruses can induce autoimmunity through erroneous bystander activation by disturbed immune surveillance. The body's defender cells usually constantly monitor and actively protect the body through immune surveillance, calling upon the proper defensive immune cells when a threat is recognized. However, a disturbance of the immune surveillance can lead to the recruitment of autoreactive Th1 and Th1/Th17 cells instead, which can result in tissue destruction.

**Figure 14.** How viruses can induce autoimmunity through epitope spreading. Persistent viral infection and continued tissue damage leads to the release of even more and more new self-antigens, increasing the number of self-tissue targets as the autoreactive T cells spread their attacks to more epitopes.

Among these mechanisms, molecular mimicry has attracted the most attention (see Figure 11). This mechanism is possible because the infecting virus shares structural similarity with one or more of the infected host's own tissue antigens, which activates B and T cells and leads to a cross-reactive immune response that attacks both the viral antigens and the host's tissue antigens [178]. Molecular mimicry has been ascribed as the mechanism for HSV-induced HSK [154], virus-induced diabetes [179], Coxsackievirus-mediated autoimmune myocarditis [180], Theiler's murine encephalomyelitis virus-induced demyelinating disease [181], and many more [182]. In fact, different viruses themselves can share molecular mimicry with each other. This can cause problems for accurate testing and diagnosis. Our own lab performed 722 serologic tests for a panel consisting of measles and herpes viruses on samples from three different clinics [183]. Several samples showed elevations for multiple viruses, with some for as many as four viruses all at the same time. If these results were to be taken at face value alone, the conclusion or diagnosis would be that the patients had multiple active infections simultaneously, including measles, even when there was no symptomatology supporting measles. Or, the more logical hypothesis would be that cross-reactivity between different viruses was generating false positives. To

investigate this, we used the NIH/US National Library of Medicine's BLAST to determine AA sequence similarity between selected viruses and HHV-6. Even when we set search cutoffs at 50% identity match and above, and places of identification to 10 places and above, we found a staggering number of HHV-6 peptide sequences that shared significant similarity with other viruses, as shown in Table 3. Only two out of the many matches for each virus are shown for brevity's sake. We believe, then, that in cases such as this of simultaneous detection of IgG and IgM antibodies against multiple infections, practitioners should remember the reality of cross-reactivity between different viruses.

**Table 3.** Potential cross-reactive epitopes between HHV-6 and other viral proteins.

| Other Viral Antigen | Other Viral Sequence | Mapped Start to End | HHV-6 Sequence | ID (%) |
|---|---|---|---|---|
| Crystal Structure of NendoU (Uridylate-specific endoribonuclease, nsp15) of SARS-CoV-2 | SHHHHHHSSG | 4–13 | SHHHHHHSSG | 100 |
| Peptide-bound SARS-CoV-2 Nsp9 RNA-replicase | HHHHHHSAAL | 3–12 | HHHHHHSSGL | 80 |
| HSV-1 portal vertex-adjacent capsid/CATC, asymmetric unit | DPPSAIPPPPPS | 347–358 | DPPRT—PPPS | 58 |
| Crystal Structure of a gE-gI/Fc complex of HSV-1 | TPPPTPADYDE | 148–158 | TPPPS—YSE | 55 |
| Atomic structure of the herpes simplex virus type 2 B-capsid | ATIAAVRGAFE | 609–619 | ATIGMVRGLFD | 64 |
| Structure of the Herpes simplex virus type 2 C-capsid with capsid-vertex-specific component | DPRPSPPTPS | 2634–2643 | DPPRTPP-PS | 60 |
| An atomic structure of the HCMV capsid with its securing layer of pp150 | KL-LVKELRMC-LS | 233–244 | KLQLDKQL—CGLS | 57 |
| Human Cytomegalovirus protease | VYVGGFLARYDQSPDE | 14–29 | VWVGGFLCVYGEEPSE | 56 |
| Epstein-Barr virus protease | GKLSFFDHVSIC | 132–143 | GK-PFFHHVSVC | 67 |
| EBV major envelope glycoprotein | SKKL-PINITAGEE | 108–120 | SKTLFPIPRSA-EE | 57 |
| Structure of Varicella-zoster virus protease | DGN-FFTHVALC | 123–133 | DGKPFFHHVSVC | 58 |
| gHgL of Varicella-zoster virus in complex with human neutralizing antibodies | TG-AI-MDIIII | 737–746 | TGLAIAM-ILFI | 58 |
| Crystal structure of measles N0-P complex | LKAEPIGS-LA | 408–417 | LTTEP-GSELA | 64 |
| Crystal structure of the prefusion form of measles virus fusion protein | DLIGQKLGLKL | 84–94 | DLL—KLNKKL | 55 |
| B. burgdorferi BmpD nucleoside binding protein bound to adenosine | LNINIIEKASTG | 78–89 | LNINHNEKATIG | 67 |
| Structure of DNA gyrase A C-terminal domain [Borrelia burgdorferi] | VIKLNDKDFV | 144–153 | VI—NDTSFV | 60 |

Only two of the many matches for each virus and only matches with identity percentages of 50% and above are shown.

The second mechanism is bystander activation (see Figure 12). This is where a non-specific over-reactive antiviral immune response causes the release of self-antigens from damaged tissue, which are taken up by antigen-presenting cells and presented to stimulate the activation of autoreactive T cells in a T-cell-receptor-independent and cytokine-dependent manner, in effect creating a localized proinflammatory environment [184]. Cytokines are the most important factors in inducing bystander activation of T cells. Bystander T cells lack specificity for the pathogen, but can still affect the immune response to the infection. In fact, antigen-independent bystander activation of T cells can either contribute to immune protection, or initiate aberrant immune responses, such as immunopathology or autoimmunity.

The third mechanism, disturbed immune surveillance, is actually an example of the aberrant immune responses that can be brought about by bystander activation (see Figure 13). The immune system has the body in a constant state of immune surveillance, in which the many parts of the immune system watch out for invading pathogens and react against them. However, it has been observed that in patients with MS, relapses can be triggered when the body's immune surveillance is disturbed, leading to erroneous bystander activation. Instead of recruiting helpful T cells, a disturbance in immune surveillance can lead to the recruitment of autoreactive Th1 and Th1/Th17$_{CM}$ cells, or even infected CXCR3$^+$ B cells capable of transporting viruses [176]. These cells can attack uninfected cells, such as the myelin sheathing of neurons, causing autoimmune tissue damage.

The fourth mechanism, epitope spreading, is related to bystander activation (Figure 14). Epitopes are the determinants on an antigen that enable it to bind with a specific antibody. Epitope spreading occurs when a viral infection triggers the release of more self-antigens and an immune response develops toward other epitopes that are separate from and non-cross-reactive with the original disease-crossing epitope [182].

The first two mechanisms have both been observed in the experimental autoimmune encephalomyelitis model of MS [185], myasthenia gravis mediated by West Nile virus [186], Theiler's murine encephalomyelitis virus-induced demyelinating disease [187], and other diseases. Viruses can also immortalize autoreactive effector cells as has been shown with EBV-infected B cells [188]. Although many mechanisms have been proposed and the evidence is substantial, the precise relationships and contributions between viruses and autoimmunity still need more investigation and greater understanding. In the interest of clarity, in Table 4 below we have summarized various studies reporting the role of HSV 1+2, HHV-6, EBV, CMV, and VZV in different autoimmune diseases with their proposed mechanisms of action. Because of the multifactorial nature of autoimmune diseases, we should remember that these viruses, and, in fact, infections, in general, are just one out of many variables that play a role in the pathophysiology of autoimmunity.

In summary, there are variable mechanisms by which herpesviruses can trigger autoimmunity. Molecular mimicry, bystander activation and epitope spreading have been reported in herpesvirus-induced autoimmunity. As neurotropic viruses, herpesviruses can directly infect and kill CNS cells, which can lead to various autoimmune disorders. The HSV viruses have been found to be involved in autoimmune and neurological disorders such as stromal keratitis and Alzheimer's disease [189–191]. VZV infections have been implicated during early pregnancy with congenital anomalies and even life-threatening infections in the newborn [192,193]. Some autoimmune phenomena, including giant cell arteritis and optic neuritis have been observed in patients with VZV infection [194,195]. EBV has been implicated in a variety of autoimmune disorders, including SSc, lupus, MS, thyroiditis, and autoimmune hepatitis [196–201]. EBV actually has the capacity to immortalize autoreactive infected B cells. Like EBV, human herpesvirus 6 (HHV-6) has been found to be involved with many autoimmune disorders, such as MS, thyroiditis, connective tissue disease, collagen vascular disease, autoimmune hemolytic anemia, and SS [202–208]. CMV is also associated with diverse autoimmune diseases, although not to the same extent with which EBV and HHV-6 have been associated with autoimmunity [209–212]. The herpesviruses also have a nasty habit of playing dead and lying dormant, resurfacing later, sometimes nastier than before in susceptible individuals. The role of herpesviruses in the pathophysiology of autoimmune diseases deserves even more scrutiny, and it is equally important to have accurate means of measuring and monitoring their appropriate biomarkers.

**Table 4.** Viruses, autoimmune diseases, and proposed mechanisms of action.

| Autoimmune Disease | Virus | Proposed Mechanisms | References |
|---|---|---|---|
| Autoimmune encephalitis | HSV | Molecular mimicry | [213] |
| Encephalitis (Human herpes encephalitis) | HSV | Molecular mimicry | [214] |
| Encephalitis and chronic neurological sequelae | HSV | Molecular mimicry? | [215] |
| Stromal keratitis | HSV | Bystander activation | [216] |
| Alzheimer's | HSV | Unknown | [217] |
| Multiple sclerosis | VZV | Unknown | [218] |
| Lupus erythematosus | EBV | Molecular mimicry | [164] |
| Autoimmune hepatitis | EBV | Molecular mimicry; Persistence of EBV in B cell | [219,220] |
| Graves' disease | EBV | EBV B-cell activation | [221] |
| Hashimoto's disease | EBV | Unknown | [222] |
| Multiple sclerosis | EBV | Molecular mimicry; Molecular mimicry; Activation of Th1, Th17, Th1/Th17 | [176,223,224] |
| Rheumatoid arthritis | EBV | Molecular mimicry | [225] |
| Sjögren's syndrome | EBV | B cell activation | [220] |
| Systemic sclerosis | CMV | Molecular mimicry; Induction of inflammation | [226,227] |
| Type 1 diabetes mellitus | CMV | Unknown | [172] |
| Systemic lupus erythematosus | CMV | Epitope spreading | [209] |
| Rheumatoid arthritis | CMV | Aggravation of inflammation | [228] |
| Endothelial cell autoimmunity | CMV | Molecular mimicry | [229] |
| Autoimmune thyroiditis | HHV-6A | NK cell killing of HHV-6 infected thyrocytes | [230] |
| Multiple sclerosis | HHV-6A/6B; HHV-6 | Infecting astrocytes and oligodendrocytes; Molecular mimicry; Activation of Th1, Th17, Th1/Th17 | [202,231,232] |
| Collagen vascular disease | HHV-6 | Molecular mimicry | [203] |
| Connective tissue disease | HHV-6 | Molecular mimicry; Selective reactivation | [204,233] |
| Sjögren's syndrome | HHV-6 | Molecular mimicry; Polyclonal activation | [205] |
| Autoimmune hemolytic anemia | HHV-6 | Molecular mimicry; Polyclonal activation | [206] |

## 5. The Role of the Gut Microbiome in the Pathophysiology of Autoimmune Diseases

We know that environmental factors such as toxic chemicals, foods and pathogens can induce an autoimmune response. But how do they do this, exactly? How do environmental factors come into contact with and penetrate into our bodies and immune systems? They do this through what we breathe, what we eat, what we drink, and what we touch, through our respiratory system, our mouth and therefore our gut, our skin, and body orifices. And as these factors encounter these entry points into our body, they come in contact and interact with our microbiome.

The microbiome is defined as the collective genomes of the microbes that populate our bodies. This includes bacteria, bacteriophage, fungi, viruses, protozoa, and helminths [234]

It is clear that our microbiome exerts significant effects on all aspects of our physiology. As man has evolved, so, too, has our microbiome, because what twenty-first century man eats, breathes and experiences is literally millennia different from what his primitive ancestors did. Blaser theorized that the losses of certain bacterial species of our ancestral microbiota have changed the conditions in which immunological, metabolic, and cognitive development occur in early life, resulting in increased disease [235]. In particular, the lifestyle changes brought about by industrialization, such as increases in sanitation, antibiotic use, consumption of processed foods, and urban living have unavoidably influenced the composition of our internal microbial communities [236]. In other words, our microbiomes are affected by our environmental factors or exposomes.

Given that we know that environmental factors can induce autoimmune responses, and that they interact with the various parts of our microbiome, it is not surprising to learn that in susceptible or genetically pre-disposed individuals, dysbiosis of the gut, oral and skin microbiome has been linked to autoinflammation and damage to the tissues. Since an altered microbial composition can induce inflammation and a breakdown in immune tolerance, changes in the human microbiome could therefore contribute significantly towards autoimmunity. In particular, we are going to focus on the gut microbiome. This is because a stable, healthy and balanced gut microbiome not only helps the body to efficiently absorb nutrients, it also helps to regulate the immune system. Thus, dysbiosis of the gut microbiome could lead to multiple autoimmune diseases [236–239].

A growing number of studies in both animal models and humans clearly demonstrate the impact of the gut microbiome on the pathogenesis of autoimmune diseases, showing convincing links between altered microbiota composition and disorders such as SLE, MS, RA, SSc, IBD, and UC [240–245].

The mechanisms for the induction of SLE, one of the most prominent autoimmune diseases, are still not fully understood, and hormonal, genetic, and environmental factors could potentially contribute towards SLE flares [246]. Recently, studies have suggested that alterations in the composition and balance of the gut microbiota may correlate with SLE disease activity. It was observed that patients with SLE had a lower Firmicutes/Bacteroides ratio and abundance of several genera. SLE patients also had reduced Lactobacillaceae and increased Lachnospiraceae [247–250]. Female SLE patients showed increases in serum sCD14, fecal secretory IgA, calprotectin levels, and *Ruminococcus gnavus* of the Lachnospiraceae family [251]. SLE patients also showed increased levels of endotoxin lipopolysaccharide (LPS), possibly due to leaky gut, which suggests that chronic microbial translocation can contribute to the development of SLE [252,253]. In lupus-prone NZBxW/F1 mice, bacterial amyloid/DNA complex was shown to stimulate autoimmune responses, including the production of type 1 IFN and autoantibodies [254]. Young lupus-prone mice showed increases in Lachnospiraceae and marked depletion of lactobacilli compared to age-matched healthy controls [255].

IBD is thought to develop as a result of interactions between environmental, microbial and immune-mediated factors in a genetically pre-disposed host. It has been shown that the gut microbiome in patients with IBD is different when compared to those in healthy control subjects. Other evidence in support of a fundamental role for the microbiome in patients with IBD includes identification of mutations in genes involved in microbiome-immune interactions among patients with IBD [256].

Boziki et al. in a 2020 review article showed that the gut microbiome was also involved in MS [257]. They focused on the gut-brain axis and how gut microbiota were essential or critically involved in both innate and adaptive immunity. They concluded by proposing that manipulation of the gut microbiota could lead to possible therapeutic protocols for MS.

The proposed mechanisms by which microbiome dysbiosis may induce autoimmune disease are shown in Figure 15.

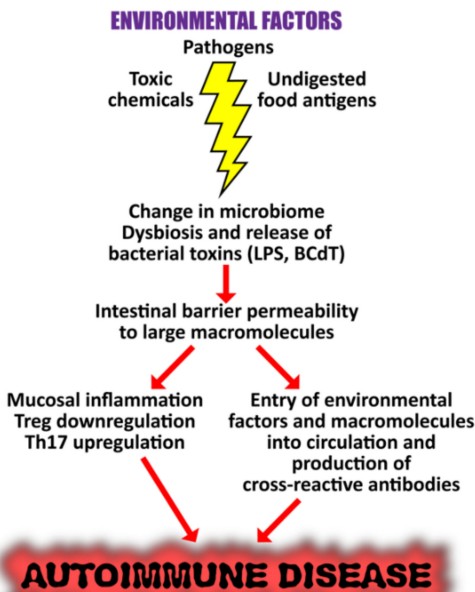

**Figure 15.** Proposed mechanisms by which microbiome dysbiosis may induce autoimmune disease. Environmental factors such as toxic chemicals, food antigens, and pathogens can affect mucosal inflammation, intestinal barrier function, and immunity, causing increased translocation of bacteria or their metabolites, such as circulating endotoxin LPS. This promotes systemic aberrant anti-inflammatory responses, eventually resulting in autoimmune disorders.

These observed changes in the microbiome caused by exposure to environmental factors are very important findings and deserve further investigation of their exact roles in the causality of autoimmune disease. Furthermore, studies show that dietary intervention with retinoic acid, prebiotics, and probiotics with the capacity to regulate Tregs can restore lactobacilli and improve symptomatology. The results demonstrated impressive changes in the gut microbiota in murine lupus and gives support to the use of retinoic acid as a dietary supplement for relieving inflammatory flares in patients with lupus [258,259]. In light of this information, characterization and manipulation of the composition of the microbiome could represent therapeutic possibilities for the improvement and possibly complete recovery of the dysfunctional immune system in different autoimmune diseases.

The microbiome, and the gut microbiome in particular, are so important to our immune system because of the microbiota that populate it. Microbiota are the diverse set of commensal bacteria that normally colonize the human body and serve as the first line of defense against infectious diseases. They modulate susceptibility to and even the severity of infections. Tan et al. propose that one must not only look at the microbiota itself, one must look at and understand how microbiota interact with the host and the pathogen, because it is the intricate interactions between these three factors that determine the outcome of the infection [260]. Understanding how microbiota affects a patient's susceptibility to the disease and the severity of the infection will facilitate the design of microbiota-based therapeutic protocols.

The growing understanding of the microbiota's roles, both beneficial and harmful, in health and disease have spurred efforts to develop disease management therapies targeting the microbiome. One such protocol that seems to be slowly rising in popularity is fecal microbial transplantation (FMT). FMT, also known as stool transplantation or bacteriotherapy, is the transfer of stool from a healthy donor into the gastrointestinal tract of a patient to change the recipient's gut microbial composition and confer a health benefit. Once used primarily to treat *Clostridium difficile* infections, it is now being investigated for possible applications in the treatment of inflammatory bowel disease, obesity, metabolic syndrome, and functional gastrointestinal disorders [261]. In one study, mice colonized

with microbiota from IBD patients developed an abundance of Th17 cells, a deficiency of Treg cells, and susceptibility to colitis [262]. Transplantation of microbiota from healthy donors led to an induction of Treg cells, reduction of Th17 cells, and protection from colitis.

Shamriz et al. point out that autoimmune diseases have a multifactorial etiology, including genetics and environmental factors [263]. They particularly stress the critical role of the microbiota in the pathogenesis of autoimmunity, and suggest that the other factors such as genetics, gender, pregnancy, and diet influence autoimmunity by affecting the composition and activity of the microbiota.

Similarly, Klag and Round found that gut microbiota, diet, immunity, and genetics interact with one another [264]. Microbiota are controlled by the immune system, but also influence the development of immunity. The immune system uses antibodies as one mechanism that directly targets microbes in an antigen-dependent manner. These antibodies are primarily IgA, although IgG and IgM have also been detected. Evidence is rapidly accumulating that IgA is essential for regulating the composition and function of the resident commensal microbiota [189,190,264]. Immune response in the form of antibodies against microbiota or against specific organs may be used to develop therapeutic interventions or as biomarkers, and Klag and Round suggest that anti-commensal antibody detection might be a novel diagnostic assay that can be used to monitor disease severity or treatment response [264].

Bishai and Palm, on the other hand, chose to look beyond the trillions of bacteria that colonize the human gut, and look instead at the thousands of microbial metabolites collectively generated by them [265]. These unique small molecules can accumulate both locally and systemically, and have the potential to affect all aspects of our biology, including our immunity, metabolism, mood, and behavior.

Technological advances in the last two decades now allow us to look even at the molecular level to examine the reciprocal interactions between host, pathogen, and microbiota. Thus, as we showed in the preceding article, we can see that different environmental factors play significant roles in the development of autoimmunity, and that clinicians should deal with the root causes of autoimmunity, not just ameliorate the symptoms.

*The Role of the Microbiome in the Pathophysiology of COVID-19*

As we have shown that the microbiome does indeed have an important and undeniable role in the development of autoimmune disease in general, we needn't go very far to find that there is compelling evidence that the microbiome is involved specifically in the pathophysiology of COVID-19 or SARS-CoV-2, the autoimmune disease. Since the pandemic itself has only been on the world stage for a couple of years, all the literature we are about to cite is relatively very recent. In 2021 Dotan et al. did a review article [266] and, based upon their conclusions, described "COVID-19 as an infectome paradigm of autoimmunity." They concluded that the evidence of the reviewed literature showed that the microbiome, particularly the gut and lung microbiome, could have a vital role in the pathogenesis, clinical severity, outcomes, and even treatment of COVID-19.

Wang et al. reviewed the microbial characteristics of COVID-19 from the aspects of the lung, gut, and oral microbiomes [267]. They showed that the composition of the microbiome of COVID-19 patients changed significantly compared to that of healthy people, particularly the lung and gut microbiota, suggesting that these microbiota can be used as biomarkers for ARDS and COVID-19.

Another important fact is that gut microbiota can restrict or inhibit NETosis, which may have a very interesting relationship with COVID-19 [268]. Neutrophils are cells that are part of our immune army, participating in both innate and adaptive immune responses through various mechanisms, one of the most important of which is the formation and release of neutrophil extracellular traps (NETs). These NETs are released during a regulated form of neutrophil cell death called NETosis. Unfortunately, although NETosis participates in the body's defense against infection, evidence indicates that it also plays an important role in the pathogenesis of several non-infectious pathological conditions, such

as autoimmune diseases and even cancer [269]. With regards to COVID-19, a crucial mechanism of the disease is the recruitment and activation of neutrophils at the infection site. Abundant NETosis and NET generation has been observed in many COVID-19 patients, resulting in unfavorable coagulopathy and immunothrombosis [270]. Additionally, excessive NETosis and NET generation are now widely recognized as mediators of additional pathophysiological abnormalities following SARS-CoV-2 infection [270].

There are many other recent publications echoing the connection between the microbiome and COVID-19, particularly the gut and lung microbiomes [271–274], and some even suggest how the microbiome can be helpful in management of the disease. What we can take away from all of this is that the connection between the microbiome and COVID-19 is very real and cannot be ignored.

## 6. Conclusions

In Part One of this two-part review, we pointed out that pathophysiological changes can occur years before the full onset of autoimmune diseases. These changes can be brought about by both internal and external, genetic and environmental factors, genome and exposome. We discussed toxic chemicals and food, two of the three most important environmental factors, in our previous article, and in this current review we have focused on the third, pathogens or infections.

It has been well established that infections are major contributors to autoimmune diseases. We have described the various mechanisms by which they accomplish this, foremost being molecular mimicry, which is when structural similarity between the pathogen and one or more human tissue antigens leads to an erroneous immune response against not just the pathogen but the human tissue as well, resulting in autoimmune reactivity, and, if not modulated, eventually full-blown autoimmune disease. We selected oral pathogens, SARS-CoV-2, and the herpesviruses as three prime examples of bacterial- or viral-induced autoimmunity.

We have shown that oral pathogens are not just limited to common infections of the mouth, but in fact have widespread impacts on areas distant from the oral cavity through various pathways and leading to autoimmune diseases, such as RA.

We have highlighted how the advent of the terrible, world-shaking COVID-19 pandemic has put a relative newcomer, SARS-CoV-2, in the spotlight as a major player, not just in epidemics in general, but in autoimmunity in particular, so much so that it has been called "the autoimmune virus." We have listed and described in detail the evidence that supports this appellation.

We have described how viruses, in particular the herpesviruses, are associated with the induction of many different autoimmune diseases, and the specific mechanisms by which they contribute to autoimmunity.

Lastly, we cannot discuss the role of pathogens in autoimmunity and focus solely on harmful viruses and bacteria without acknowledging the microbiome and the commensal bacteria or beneficial microbiota that populate it. The microbiota modulates disease susceptibility and severity, but we have seen the proposition that it is the three-way reciprocal interaction of host–pathogen–microbiota that determines the outcome of the disease. In particular, we have looked at the role of the gut microbiome in the pathophysiology of autoimmune diseases, because the gut microbiome has an important role in regulating the immune system. Environmental factors such as food, toxic chemicals, and pathogens, including oral bacteria, can change the composition of the gut microbiome, causing the release of toxins and resulting in leaky gut. Consequently, disruptions and imbalances of the gut microbiome can lead to susceptibility to or induction of multiple autoimmune disorders.

The most challenging aspect of understanding and treating autoimmunity has always been finding the root causes of these diseases, and that means identifying early events that trigger the sometimes years-long process of pathogenesis. Environmental factors account for up to 70%, but a dysfunctional gut microbiome can also play a critical role in disrupting

the immune system [275], and this dysfunction can itself be caused by environmental factors. The role of the exposome and early triggering events are shown in Figure 16.

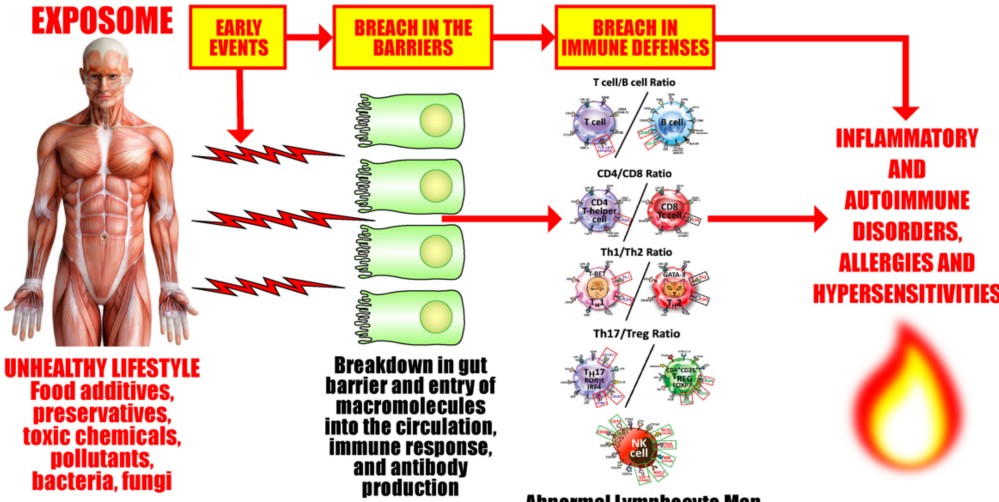

**Figure 16.** Exposome factors and early events that are involved in the breach of barriers and immune defenses that are associated with inflammatory, autoimmune, neurodegenerative disorders, allergies, and hypersensitivities.

Another challenge regarding autoimmunity is detecting it, identifying it, and measuring it. Since by now we have been talking for two articles about the huge role of the exposome in the pathophysiology of autoimmunity, then we know that we are basically talking about biomarkers for the exposome. As has been said before, the exposome is the sum of your lifetime exposure to external and internal environmental factors. Food, chemicals, infections, lifestyle, and habits all contribute to it. An article by Wallace at al. proposes that the factors that make up an exposome can be measured by biomarkers in blood, breath, and urine, and proposes a new name for them: bioindicators [276]. But here lies the challenge. There is no one biomarker to measure your exposome, or even ten. With new ones constantly either being discovered or developed, there are countless foods, infections, viruses, and chemicals out there, and it is a gargantuan task to find ways to find accurate biomarkers for them that we can use to improve and maintain our health.

Understanding the precise roles and intricate relationships shared by the exposome and all these factors, internal or external, genetic, or environmental and finding the early events and root causes of these disorders can help us to develop better strategies and therapeutic protocols for the management of the epidemic of autoimmunity that has emerged to plague the modern world.

**Author Contributions:** A.V., E.V., A.Z.R. and Y.S.: writing—original draft preparation; writing—review and editing. All authors have read and agreed to the published version of the manuscript.

**Funding:** This research received no external funding.

**Institutional Review Board Statement:** Not applicable.

**Data Availability Statement:** Not applicable.

**Acknowledgments:** The authors wish to thank Joel Bautista for the preparation of this manuscript for publication.

**Conflicts of Interest:** A.V. is co-owner, CEO, and technical director of Immunosciences Lab., Inc. E.V. is the owner and founder of Regenera Medical. A.Z.R. and Y.S. declare no conflict of interest.

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
