# Peer review of "The Role of Exposomes in the Pathophysiology of Autoimmune Diseases II: Pathogens"

_pathophysiology, doi:10.3390/pathophysiology29020020_

Round 1

Reviewer 1 Report

Review Exposome in the pathophysiology of autoimmune diseases

This profound review is well written and covers an important issue to understand autoimmune diseases. The exposome and the interaction with the host immune system is so far underrecognized and important. The review provides an overview about current concepts and the scientific rationales to address the exposome. I have only few remarks, but in general, the manuscript is informative by discussing all important aspects.

Peptidyl-arginine deaminase is very important in rheumatoid arthritis. The authors should provide a reference for the statement that this enzyme is important for inflammation outside the joints.

In the paragraph starting at line 409 and discussing T cell abnormalities in COVID 19 infection, the authors described that abnormal lymphocyte subsets were able to predict mortality. Given the previous statement given by the authors concerning the role of a Treg deficiency in autoimmunity, the authors should give some information about the ratios between Treg and Tcon or Treg and Th17 subsets in COVID-19 patients.

Concerning key point #5 starting at line 549: How did the authors show cross-reactivities? Normally, this would require purification of specific abs. Were the cross-reactive abs of the COVID-19 induced human abs the same recognized by the monoclonal commercially available abs? This would be a good validation.

In the same section, the authors describe a study very recently published in Nature comm. In this manuscript, I am not aware of any study analysing cross reactivities or IgA abs to tissue autoantigens. The authors should provide the correct reference.

The authors should provide references for their statement that cross-reactivity was different between patients with severe or mild COVID-19 infection.  Which manuscript shows these data?

The text between Figure 9 and 10 is missing. The sentence starting “Finally, viral infection or reactivation with EBV… has no end and probably the paragraph is missing. Figure 10 should be described.

The authors described four mechanisms important for autoimmunity by Herpesviruses. However, first, they should introduce these four mechanisms briefly starting the discussion. In addition, in table 4, they should refer to these four mechanisms to clarify the proposed mechanisms to be consistent in the manuscript.

Concerning the role of the microbiome, one strong argument would be the change of a disease or of a phenotype by fecal transplantation. The authors should discuss this and should provide data in addition to associations and correlations.

The authors should provide the reference for the statement that Treg can restore the microbiome.

The identification of biomarkers for the exposome remains a challenge I would recommend to discuss this issue.    

Author Response

Thank you for your help and recommendations. We have attached/submitted a word file with our answers to your comments  in red font. We hope the manuscript is now suitable for publication.

Reviewer 2 Report

The manuscript submitted by Vojdani et al., titled: "The Role of Exposomes in the Pathophysiology of Autoimmune Diseases II: Pathogens" is an extensive review that discusses how exposures can influence outcomes of autoimmune diseases.

This is an interesting topic with several implications at the public health and clinical levels. The manuscript is well written and organized providing a wealth of information.

The reviewer would like to raise the following conceptual point for the improvement of the manuscript.

The manuscript minimally points and discusses the diet as an exposure factor. It is well established however that there are certain nutrients and/or foods that are significantly implicated in autoimmune diseases as well as chronic diseases in general. While the reviewer understands that this is not the focus of the paper he believes that a section where diet would be briefly discussed in the context of nutrient extending signaling and modulation of risk for metabolic and other diseases may be of interest and would improve the manuscript. Along those lines it is also important to point out that there are exposures that may be beneficial towards disease risk (hence reducing disease risk). Medical Nutrition Therapy essentially is an entire field working with this premise. 

Here are a couple of manuscripts that may be helpful towards this aspect:

Kristo, A.S.; Klimis-Zacas, D.; Sikalidis, A.K. Protective Role of Dietary Berries in Cancer. Antioxidants 20165, 37. https://doi.org/10.3390/antiox5040037 

Sikalidis, A.K. Amino Acids and Immune Response: A Role for Cysteine, Glutamine, Phenylalanine, Tryptophan and Arginine in T-cell Function and Cancer? Pathol. Oncol. Res. 21, 9–17 (2015). https://doi.org/10.1007/s12253-014-9860-0 

Author Response

(The authors gave the same response as above.)

Reviewer 3 Report

The paper is interesting and very comprehensive.

Please add P. gingivalis in table 1.

Please add a brief sentence on the concept of infectome (Infectome: a platform to trace infectious triggers of autoimmunity. Bogdanos DP, Smyk DS, Invernizzi P, Rigopoulou EI, Blank M, Pouria S, Shoenfeld Y.Autoimmun Rev. 2013 May;12(7):726-40. doi: 10.1016/j.autrev.2012.12.005. Epub 2012 Dec 22)

Author Response

(The authors gave the same response as above.)

Reviewer 4 Report

This review -a second article of a series of two- focuses its content on the role of exposomes in autoimmune diseases with a special focal point on the importance of pathogens. It describes various molecular and cellular mechanisms that are central in this process with three examples of pathogens that illustrate how they occur, an oral bacterium (P. gingivalis) and two viruses (SARS-CoV-2 and herpesviruses). The authors extensively describe the mechanisms by which these pathogens could contribute to autoimmunity. At the end, they look at the role of gut microbiome in the pathophysiology of autoimmune disorders. Overall, the review gives a comprehensive description of the problem addressed; the available scientific data are well incorporated in the text. The general topic discussed in this review is both important and timely. This article is well written, the different topics are clearly presented and discussed, reporting extensive examples and citations. The figures and tables are comprehensive and helpful. The present work needs minor revisions.

1. I suggest ameliorating the division of chapters for a better distinction of single infectious agents (1. Introduction, 2. Oral pathogens, 3. SARS-CoV-2, 4. Herpesvirus....).

2. Abbreviations: Some acronyms might be written in extended form when are mentioned for the first time. Table 1 contains several abbreviations (HHV, EBV, HLA, SARS-CoV-2) that should be described in the legend. GAD-65 (line 100), CBC (line 305), COPD (line 329), MRI (line 352), CRP (line 407), ICU (line 408), MOG (line 666), AA (line 691), TH1/Th17CM (line 731) FOXP3, T-BET and RORgT (figures 10 and 13), TMEV-IDD (line 757), BCdT (figure 15), ENA (line 516), ARDS (line 917), and others are not described. CD is used for celiac disease (page 4) and cluster of differentiation (page 8). Certain abbreviations are described several times in the text (e.g., RA, in lines 60, 365, 619; also for MS, SLE, SS, EBV, BLAST, VZV, etc.). UC is included in IBD (page 22, line 826). Certain abbreviations are given but they are not used later (e.g., ADs line 47, WBCs). Line 709, use APCs.

3. Adjust the style of some references (ref. 72,73, from 172 to 194).

4. Line 106: When the authors refer to epitope spreading (an immune feature that can be not well known to the readers), they could at minima refer to Authors like Craft J & Fatenejad S (1997) https://doi.org/10.1002/art.1780400803; James JA & Harley JB. (1998) doi: 10.1111/j.1600-065x.1998.tb01220.x.; Sercarz EE (2000) doi: 10.1006/jaut.2000.0380 who published key seminal findings in this field with concrete development (e.g., Monneaux F, Muller S. Epitope spreading in systemic lupus erythematosus: identification of triggering peptide sequences. Arthritis Rheum. 2002 Jun;46(6):1430-8. doi: 10.1002/art.10263).

5. Line 114: actually gum disease can be induced by many other bacteria, often acting as a biofilm; the main periodontal pathogens are P. gingivalis, Treponema denticola, Fusobacterium nucleatum. See Gao, L. et al. Polymicrobial periodontal disease triggers a wide radius of effect and unique virome. NPJ Biofilms Microbiomes 6, 10 (2020) and many other articles. This remark does not call into question the importance of P. gingivalis.

6. A few typo. Line 221, delete will; line 257, replace COVID-10 by COVID-19; figure 14 multiple; line 906 is unclear.

7. A few expressions could be improved: line 263: “terrible disease”; line 288: the host body is “in good health”; and cells are “in harmony”.  Line 324 and next sentences, Th1 and Th17 cells (“cells” missing). Some sentences are repetitive and could be deleted or shortened to make the text more fluid. Line 399, COVID-19 should be SARS-CoV-2.

8. Line 459: the term “proof” should be “argument” since it has not yet been experimentally proven. Cross-reactivity between monoclonal antibodies generated after immunizing mice or rabbits with SARS-CoV-2 and self-components is less relevant than human monoclonal Abs that are produced without immunisation in adjuvant (Freund’s adjuvant is known to open the BBB, for example, and induced hyperimmunisation). This may be better highlighted in line 503 and the next sentences as it is important in the reasoning in this important paragraph (key point No 3).

9. The Authors may wish to better define homology and identity.  

10. A legend could be added to the Table 3.  

11. The term “exposome” (in the title) is used for the 1st time in line 813 (page 21) of the text. The focus on exposome, which is effectively very important, could be better and earlier highlighted in this article.

12. Section 3: while the role of microbiome is important in SLE, a few sentences in other ADs could be added, especially in IBDs, periodontal diseases and others.

13. An important aspect that has not be addressed in this review is the relation between NETosis (crucial in ADs) and infectious agents, especially SARS-CoV-2.

Author Response

Thank you for your help and recommendations. We have attached/submitted a word file with our answers to your comments  in red font. We hope the manuscript is now suitable for publication.

This manuscript is a resubmission of an earlier submission. The following is a list of the peer review reports and author responses from that submission.